# BOUNDS ON THE RECONSTRUCTION ERROR OF KERNEL PCA WITH INTERPOLATION SPACES NORMS

## ABSTRACT

In this paper, we utilize the interpolation space norm to understand and fill the gaps in some recent works on the reconstruction error of the kernel PCA. After rigorously proving a simple but fundamental claim appeared in the kernel PCA literature, we provide upper bound and lower bound of the reconstruction error of the empirical kernel PCA with interpolation space norms under the assumption $(C)$, a condition which is taken for granted in the existing works. Furthermore, we show that the assumption $(C)$ holds in two most interesting settings (the polynomial-eigenvalue decayed kernels in fixed dimension domain and the inner product kernel on large dimensional sphere $\mathbb{S}^{d-1}$ where $n \asymp d^\gamma$) and compare our bound with the existing results. This work not only fills the gaps appeared in literature, but also derives an explicit lower bound on the sample size to guarantee that the (optimal) reconstruction error is well approximated by the empirical reconstruction error. Finally, our results reveal that the RKHS norm is not a relevant error metric in the large dimensional settings.

## 1 INTRODUCTION

Principal Component Analysis (PCA), a widely used statistical technique for dimensionality reduction and data visualization, aims at finding a subspace of dimension $\ell$ such that the data after projection retaining as much of the original variance as possible (Jolliffe, 2002). It is easily seen that the subspace is spanned by the $\ell$ eigenvectors corresponding to the first $\ell$ largest eigenvalues of the covariance matrix. In practice, if we observed that $\mathbf{X} = (X_1, X_2, \cdots, X_n)$ are i.i.d sampled from a distribution $P$ on $\mathcal{X} \subseteq \mathbf{R}^d$, we may use the largest eigenvectors of the empirical covariance matrix $\frac{1}{n} \sum_{i=1}^n X_i X_i^\mathsf{T}$ to produce estimates of the first $\ell$ eigenvectors.

PCA works well when the relationships between variables in the data are approximately linear. Kernel PCA, on the other hand, is a non-linear dimensionality reduction technique which allows for capturing non-linear relationships in the data. For a reproducing kernel Hilbert space (RKHS) $\mathcal{H}$ associated with the kernel function $k : \mathcal{X} \times \mathcal{X} \to \mathbf{R}$, the kernel PCA would produce a subspace spanned by the eigenvectors corresponding to the $\ell$ largest eigenvalues of the covariance operator $\Sigma : \mathcal{H} \to \mathcal{H}$ defined as

$$\Sigma f = \mathbb{E}_{X \sim P}[\Phi(X) \otimes_\mathcal{H} \Phi(X)](f) = \mathbb{E}_{X \sim P}[\Phi(X) f(X)],$$

where $\Phi(X) := k(X, \cdot) \in \mathcal{H}$ is called the feature map. Similarly, given $n$ i.i.d. samples $\mathbf{X} = (X_1, X_2, \cdots, X_n)$, the kernel PCA produces a subspace spanned by the $\ell$ largest eigenfunctions of the empirical covariance operator $\widehat{\Sigma} f = \frac{1}{n} \sum_{i=1}^n \Phi(X_i) f(X_i)$.

The nonlinearity of the feature map $\Phi(\cdot)$ allows kernel PCA to capture more complex data patterns than PCA. Consequently, kernel PCA has much more broad and successful applications including image denoising (Mika et al., 1998; Jade et al., 2003; Teixeira et al., 2008; Phophalia & Mitra, 2017), computer vision (Lampert et al., 2009; Peter et al., 2019), image/systems modeling (Kim et al., 2005; Li et al., 2015), feature extraction (Chang & Wu, 2015), and novelty/fault detection (Hoffmann, 2007; Samuel & Cao, 2016; de Moura & de Seixas, 2017).

However, the statistical properties of kernel PCA have not yet been well understood, especially on the convergence rate of the reconstruction error of kernel PCA. In contrast, motivated by the successful applications of neural networks and the seminal neural tangent kernel theory (Jacot et al., 2018), lots of research have been done on other types of kernel-related algorithms, especially kernel regressions

and kernel classifications. Various new problems including the minimax rate on the excess risk of the kernel regression in fixed dimensions (Caponnetto, 2006; Caponnetto & De Vito, 2007; Raskutti et al., 2014; Lin et al., 2020), the generalization performance of kernel interpolation (Rakhlin & Zhai, 2019; Beaglehole et al., 2022; Buchholz, 2022; Lai et al., 2023; Li et al., 2023b), and learning curves of kernel regression (Bordelon et al., 2020; Cui et al., 2021; Jin et al., 2021; Li et al., 2023a) make kernel regression an active research field. Therefore, it is natural to ask similar questions about kernel PCA as about kernel regressions.

Analyzing large dimensional data (e.g., $n \asymp d^\gamma$) has long been an important task in statistics and machine learning (Donoho et al., 2000). Practical data, such as financial data and modern machine learning datasets, often have dimensions ranging from thousands to millions. Thus, researchers are more interested in the performance of algorithms in large dimensional data. Unfortunately, to the best of our knowledge, no works have touched on the statistical properties of large dimensional kernel PCA. On the contrary, results for large dimensional kernel regression are fruitful. For large dimensional kernel regression, common assumptions on the eigenvalues of the kernel (e.g., the polynomial eigendecay assumption and the embedding index assumption in Li et al. (2023a); Zhang et al. (2023)) no longer hold, making the analysis more complicated. Early works (Ghorbani et al., 2021; Donhauser et al., 2021; Mei et al., 2022; Xiao et al., 2022; Misiakiewicz, 2022; Hu & Lu, 2022) discussed the polynomial approximation barrier for large dimensional kernel ridge regression concerning square-integrable function classes. Then, Lu et al. (2023); Zhang et al. (2024a) determined the convergence rate on the excess risk and the minimax optimality of kernel regression and reported several new phenomena exhibited in large dimensional kernel regression, e.g., the periodic plateau behavior. For kernel interpolation in large dimensions, Liang & Rakhlin (2020); Liang et al. (2020); Aerni et al. (2022); Barzilai & Shamir (2023) showed that kernel interpolation can generalize for specific function classes. The above new phenomena exhibited in large dimensional kernel regression bring an interesting question: Does there exist new phenomena occurring in large dimensional kernel PCA?

## 1.1 RELATED WORKS

**Reconstruction error of PCA.** PCA is commonly derived by minimizing the reconstruction error over all orthonormal basis of a $\ell$-dimensional subspace of $\mathbf{R}^d$, and it is a well-known result that the $\ell$ largest eigenvectors of the covariance matrix minimize the reconstruction error (Jolliffe, 2002). When the empirical PCA is used to estimate the principal components of PCA, one of the quantities researchers are interested in is then the reconstruction error for the empirical PCA. Bounds on the reconstruction error for the empirical PCA are derived by Shawe-Taylor et al. (2002; 2005); Blanchard et al. (2007). Under certain conditions, when $\ell \leq cn$ for a certain constant $c$, Reiss & Wahl (2020) showed that the expectation of the reconstruction error for the empirical PCA can be upper bounded by the reconstruction error for the PCA up to a constant factor. Moreover, they consider several decaying rates for the eigenvalues of the covariance matrix, and determine the (minimax optimal) convergence rate of the reconstruction error for the empirical PCA.

**Reconstruction error of kernel PCA with Hilbert space norm.** Though kernel PCA is a popular variant of PCA, the statistical properties of kernel PCA (and its empirical version) received little discussion. Early works in kernel PCA mainly considered the reconstruction error with Hilbert space norm and aimed at bounding the difference between the reconstruction error of kernel PCA and empirical kernel PCA. For example, Shawe-Taylor et al. (2005) bounded the difference between the reconstruction errors with the eigenvalues of the kernel matrix $k(\mathbf{X}, \mathbf{X})/n$; Blanchard et al. (2007) modified the bound given in Shawe-Taylor et al. (2005); Rudi et al. (2013) claimed a bound on the difference between the reconstruction errors merely based on the eigenvalues of the kernel, despite that their proof has several gaps (see Remark 3.2 for details).

**Reconstruction error of kernel PCA with $L_2(\mathcal{X}, P)$ norm.** Recently, Sriperumbudur & Sterge (2022); Sterge & Sriperumbudur (2022) considered the reconstruction error with $L_2(\mathcal{X}, P)$ norm rather than Hilbert space norm, and claimed that they have determined the convergence rate of the reconstruction error of empirical kernel PCA and variants of kernel PCA in fixed dimensions. They argued that the reconstruction error with $L_2(\mathcal{X}, P)$ norm can be generalized to several variants of kernel PCA, including the random-feature kernel PCA and the Nyström kernel PCA, while the reconstruction error with Hilbert space norm can not. However, there exist several gaps in their proof (see Remark 3.2 for details).

Table 1: Comparison on bounds of the reconstruction error of empirical kernel PCA

|  | RW20 | SS22 | Our result |
|---|---|---|---|
| Parameter of the interpolation space | $s = 1$ | $s = 0$ | $0 \leq s \leq 1$ |
| Order of upper bound | $\sum_{i \geq \ell+1} \lambda_i$ | $\mathcal{N}_\Sigma(t)(\lambda_{\ell+1} + t)^2$ | $\mathcal{N}_\Sigma(\lambda_{\ell+1})\lambda_{\ell+1}^{2-s}$ |
| Lower bound | $\sum_{i \geq \ell+1} \lambda_i$ | $\sum_{i \geq \ell+1} \lambda_i^2$ | $\sum_{i \geq \ell+1} \lambda_i^{2-s}$ |
| Polynomial eigendecay | $\ell^{-\beta+1}$ | $\ell^{-2\beta+1}$ | $\ell^{-(2-s)\beta+1}$ |
| Large dimension (hypersphere setting) | \ | \ | $d^{-(q+1)(1-s)}$ |

Comparison between our results and the results in RW20 (Reiss & Wahl, 2020) and SS22 (Sriperumbudur & Sterge, 2022) under certain conditions. The proofs of the results in SS22 contain gaps, hence we present them in grey. $\mathcal{N}_\Sigma(t)$ is a coefficient which can be bounded for no more than $O(1/n)$.

## 1.2 OUR CONTRIBUTIONS

The major contributions of the paper are as follows. Also, we provide a comparison between our results and some existing results in Table 1 for the sake of convenience.

**Upper and lower bounds on the reconstruction error of empirical kernel PCA under the interpolation space norm.** In this paper, we consider the interpolation space $[\mathcal{H}]^s$ (defined in Section 2.4) with parameter $s \geq 0$, and we introduce the reconstruction error of kernel PCA under $[\mathcal{H}]^s$ norm.

i). We develop a new technique and provide a rigorous proof of the optimality of kernel PCA with $[\mathcal{H}]^s$ norm. As a direct result, we provide a lower bound of the reconstruction error of empirical kernel PCA (Theorem 2.5).

ii). We provide an upper bound of the reconstruction error of empirical kernel PCA (Proposition 3.1). Moreover, we notice that the reconstruction error with $[\mathcal{H}]^s$ norm links the two types of reconstruction errors, i.e. the one with $[\mathcal{H}]^1 = \mathcal{H}$-norm and the one with $[\mathcal{H}]^0 = L_2(\mathcal{X}, P)$-norm. As a consequence, we could compare our results with existing results about the $\mathcal{H}$-norm in Shawe-Taylor et al. (2005); Blanchard et al. (2007); Reiss & Wahl (2020).

iii). We apply our bounds to the polynomially eigendecay kernels (i.e., the eigenvalues of the kernel satisfy $\lambda_j \asymp j^{-\beta}$ for $\beta > 1$), and we successfully determine the tight convergence rate on the reconstruction error of kernel PCA for any $0 \leq s \leq 1$ (Corollary 3.4). This type of results is often referred as the optimality of the empirical kernel PCA (e.g., Sriperumbudur & Sterge (2022); Sterge et al. (2020)). Our results not only provide a rigorous proof of the claims for $0 \leq s \leq 1$ in Sriperumbudur & Sterge (2022), but also are in accordance with the results in Reiss & Wahl (2020) when $s = 1$.

**Convergence rate of empirical kernel PCA in large dimensions under the hypersphere setting** The most interesting part of this paper is trying to see the performance of empirical kernel PCA, especially for the large dimensional data where the number of samples $n \asymp d^\gamma$ under the hypersphere setting. With the help of Proposition 3.1, we show that for a reasonable range of $\ell$ (which is characterized by a quantity $q$ introduced in Theorem 3.8), both the upper bound and the lower bound of the reconstruction error of empirical kernel PCA are of the rate $d^{-(q+1)(1-s)}$, and hence determine the optimal convergence rate in the large dimension situation.

Our results reveal two interesting phenomena only occurring in large dimensional kernel PCA. (i) We find that the reconstruction error of large dimensional empirical kernel PCA with $\mathcal{H}$-norm, which is deduced from the reconstruction error of PCA (see, e.g., Shawe-Taylor et al. (2005); Blanchard et al. (2007); Reiss & Wahl (2020)), is of order $\Theta(1)$. Therefore, we conclude that $\mathcal{H}$-norm is inappropriate for the reconstruction error of kernel PCA when considering the large dimension case. (ii) The second phenomenon is the periodic plateau behavior, and as shown in Figure 1(c), when $\ell \asymp d^\zeta$ for $\zeta \in (p, p + 1)$ with any integer $p \geq 0$, the convergence rate of the reconstruction error of (empirical) kernel PCA does not change when $\zeta$ varies. Interestingly, we find that similar periodic plateau behavior on the curve of the excess risk exists on large dimensional kernel regression. For example, Lu et al. (2023); Zhang et al. (2024a) found that the convergence rate of the excess risk of kernel regression does not change when $\gamma$ varies within certain ranges. Therefore, we believe that the periodic plateau behavior is widely exhibited in large dimensional kernel-related algorithms.

We provide a graphical illustration of the theoretical results of our work in Figure 1. The experiment part can be found in Section 4.

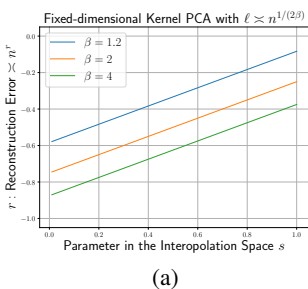 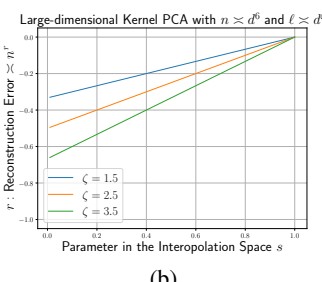 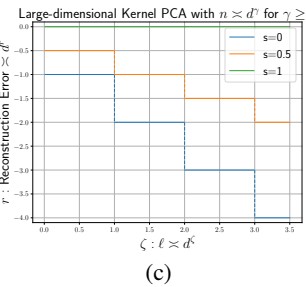

(a) (b) (c)

Figure 1: Figure 1(a) and Figure 1(b) illustrate the convergence rate on the reconstruction error of (empirical) kernel PCA with different source condition $0 \leq s \leq 1$ in (i) fixed dimensional setting and (ii) large dimensional setting. Figure 1(c) illustrate the relation between the reconstruction error and $\ell$, with $s \in \{0, 0.5, 1\}$. In all three subfigures, we use solid lines when the convergence rate of both empirical and population reconstruction error is the same, and we use dashed lines when the empirical and population reconstruction error ranges from $d^{(\zeta-1)(1-s)}$ to $d^{\zeta(1-s)}$ for any $\zeta \in \mathbb{N}$.

## 2 PRELIMINARIES

In this section, we provide a brief review of preliminary results on PCA and kernel PCA.

**Notations** Let $\mathcal{X} \subseteq \mathbf{R}^d$ be the sampling space and let the underlying probability distribution of the sampling be $P$. $\mathbf{X} = (X_1, \ldots, X_n) \subseteq \mathcal{X}$ is the set of observations under probability distribution $P$.

For a Hilbert space $\mathcal{H}$, We denote different norms as follows. $\| \cdot \|_{\mathcal{L}^1(\mathcal{H})}$ is the trace norm of an operator, $\| \cdot \|_{\mathcal{L}^2(\mathcal{H})}$ is the Hilbert-Schmidt norm, $\| \cdot \|_{\mathcal{L}^\infty(\mathcal{H})}$ is the operator norm, $\| \cdot \|_2$ is the 2-norm of $\mathbf{R}^d$, and $\| \cdot \|_{L_2(\mathcal{X}, P)}$ is the norm in the $L_2(\mathcal{X}, P)$ function space. Also, $a \otimes_{\mathcal{H}} a = \langle a, \cdot \rangle_{\mathcal{H}} a$, where $\langle \cdot, \cdot \rangle_{\mathcal{H}}$ means the inner product in space $\mathcal{H}$.

In the large-dimension setting, we consider the following asymptotic framework: We assume there exist three positive constants $c_1$, $c_2$ and $\gamma$, which satisfies $c_1 d^\gamma \leq n \leq c_2 d^\gamma$. Also, we define the following notations: $b \gtrsim a$ if and only if there exists a constant $\mathcal{C}$ only depending on $c_1, c_2, \gamma$ such that $\mathcal{C}a \leq b$. $b \lesssim a$ if and only if there exists a constant $\mathcal{C}$ only depending on $c_1, c_2, \gamma$ such that $\mathcal{C}b \leq a$. $a \asymp b$ if and only if $b \gtrsim a$ and $b \lesssim a$.

### 2.1 PRINCIPAL COMPONENT ANALYSIS (PCA)

The traditional PCA method aims at how to reduce the dimension of the data without abandoning much information (Jolliffe, 2002). Denote the diagonalization of the covariance matrix as

$$\mathbb{E}_{X \sim P} X X^\mathsf{T} = \sum_{i=1}^d \theta_i \alpha_i \alpha_i^\mathsf{T}. \tag{1}$$

where $\theta_i \in \mathbf{R}, \alpha_i \in \mathbf{R}^d, i = 1, 2, \cdots, d$ are the eigenvalues and eigenvectors satisfying that $\{\theta_i, i = 1, 2, \cdots, d\}$ is non-increasing. The method chooses the subspace spanned by the first $\ell$ eigenvectors, where $\ell$ is the goal dimension.

Similarly, the empirical covariance matrix can be diagonalized as $\frac{1}{n} \sum_{i=1}^n X_i X_i^\mathsf{T} = \sum_{i=1}^d \widehat{\theta}_i \widehat{\alpha}_i \widehat{\alpha}_i^\mathsf{T}$, where $\widehat{\theta}_i \in \mathbf{R}, \widehat{\alpha}_i \in \mathbf{R}^d, i = 1, 2, \cdots, d$ are the eigenvalues and eigenvectors satisfying that $\{\widehat{\theta}_i, i = 1, 2, \cdots, d\}$ is non-increasing. The space spanned by the first $\ell$ eigenvectors $\widehat{\alpha}_1, \cdots, \widehat{\alpha}_\ell$ can be used to approximate $\text{span}\{\alpha_1, \cdots, \alpha_\ell\}$.

For $(\beta_1, \cdots, \beta_\ell)$ as an orthonormal basis of a $\ell$-dimensional subspace of $\mathbf{R}^d$, the reconstruction error used in PCA is defined as

$$R(\beta_1, \cdots, \beta_\ell) := \mathbb{E}_{X \sim P} \left\| X - \sum_{i=1}^{\ell} (X^{\mathsf{T}} \beta_i) \beta_i \right\|_2^2. \tag{2}$$

The following result shows that the first $\ell$ eigenvectors minimize the reconstruction error in (2).

**Proposition 2.1.** *(Jolliffe, 2002) Let $\alpha_1, \cdots, \alpha_\ell$ be the eigenvectors in (1), then*

$$R(\alpha_1, \cdots, \alpha_\ell) = \min_{\substack{(\beta_1, \cdots, \beta_\ell) \\ \text{is orthonormal}}} R(\beta_1, \cdots, \beta_\ell) = \sum_{j \geq \ell+1} \theta_j.$$

From Proposition 2.1, the $\ell$ leading eigenvectors are proved to be the optimal point. Hence, one of the quantities researchers are interested in is the reconstruction error of empirical PCA, $R(\widehat{\alpha}_1, \cdots, \widehat{\alpha}_\ell)$. Reiss & Wahl (2020) gave a tight upper bound on $R(\widehat{\alpha}_1, \cdots, \widehat{\alpha}_\ell)$ which we briefly reviewed below.

**Proposition 2.2.** *(Reiss & Wahl, 2020) Suppose $X$ is sub-Gaussian. If for all $s \leq \ell$, $\frac{\lambda_s}{\lambda_s - \lambda_{\ell+1}} \sum_{j \leq s} \frac{\lambda_j}{\lambda_j - \lambda_{\ell+1}} \leq n/\left(16 C_3^2\right)$ holds, then we have*

$$R(\widehat{\alpha}_1, \cdots, \widehat{\alpha}_\ell) \leq C \sum_{j \geq \ell+1} \theta_j + C\Delta_n, \tag{3}$$

*where $\Delta_n := \sum_{i=1}^{d} \theta_i \cdot e^{-n(\theta_\ell - \theta_{\ell+1})^2/(4C'\theta_\ell)^2}$ is an exponentially small remainder term, and the constants are defined as in Theorem 2.12 in Reiss & Wahl (2020).*

*Remark* 2.3. One can easily extend results in Proposition 2.2 to kernel PCA: we only need to replace $X$ with $\Phi(X)$, the feature map of the RKHS. Notice that such replacement corresponds to a reconstruction error of kernel PCA with $\mathcal{H}$ norm (the RKHS norm). We will provide a comparison between Proposition 2.2 and our results in the next section.

Reiss & Wahl (2020) provides an upper bound of the excess risk $\mathcal{E}_\ell^{PCA} \triangleq R(\widehat{\alpha}_1, \cdots, \widehat{\alpha}_\ell) - R(\alpha_1, \cdots, \alpha_\ell)$, which turns out to be minimax optimal when the covariance operator/matrix restricted to spiked models (Vu & Lei, 2012). The Proposition 2.2 in certain situation is a standard oracle inequality with an exponentially small remainder term.

## 2.2 REPRODUCING KERNEL HILBERT SPACE

Throughout the paper, we denote $\mathcal{H}$ as a separable RKHS on $\mathcal{X}$ with respect to a continuous kernel function $k$ satisfying $\sup_{x \in \mathcal{X}} k(x, x) \leq \kappa^2$. For detailed explanation and properties of RKHS, readers may refer to Caponnetto & De Vito (2007).

Denote the inclusion map by $\mathcal{J} : \mathcal{H} \to L_2(\mathcal{X}, P)$, and the adjoint operator by $\mathcal{J}^* : L_2(\mathcal{X}, P) \to \mathcal{H}$. Consider the following operator $\Sigma : \mathcal{H} \to \mathcal{H}$, $(\Sigma f)(x) = \int_{\mathcal{X}} k(x, y) f(y) dP(y)$. Clearly, $\Sigma = \mathcal{J}^* \mathcal{J}$, and hence that $\Sigma$ is self-adjoint, positive, trace-class, and compact. Thus, by the Mercer's decomposition (Reed & Simon, 1980), we have

$$\Sigma = \sum_{i \in N} \lambda_i \langle \cdot, \phi_i \rangle_{\mathcal{H}} \phi_i. \tag{4}$$

where $N$ is an at most countable set, $\{\lambda_i, \ i \in N\}$ is non-increasing and summable, $\{\phi_i, \ i \in N\}$ are the corresponding orthonormal eigenfunctions. The results similar to the above analysis can also be found in other kernel related literature, see, e.g., Rosasco et al. (2010); Shawe-Taylor et al. (2005); Sriperumbudur & Sterge (2022).

## 2.3 KERNEL PRINCIPAL COMPONENT ANALYSIS (KERNEL PCA)

The PCA method performs well when the relationships between variables in the data are approximately linear. When the approximate linearity violated mildly, a common approach is to project the data to a higher-dimensional space $\mathcal{H}$, and then operate PCA in $\mathcal{H}$, which is known as the kernel principal component analysis (Schölkopf et al., 1998).

Specifically, for any kernel $k$, the kernel PCA method projects the data $X \in \mathbf{R}^d$ into $k(X, \cdot) \in \mathcal{H}$, and then chooses the subspace spanned by the first $\ell$ eigenfunctions of the operator $\Sigma = \mathbb{E}_{X \sim P}[k(X, \cdot) \otimes_{\mathcal{H}} k(X, \cdot)]$. From the Mercer's decomposition (4), we know that the subspace is spanned by $\phi_1, \cdots, \phi_\ell$.

The empirical kernel PCA considers the empirical version of the operator $\widehat{\Sigma} : \mathcal{H} \to \mathcal{H}$, $\widehat{\Sigma} f = \frac{1}{n} \sum_{i=1}^{n} k(\cdot, X_i) f(X_i)$. Since $\widehat{\Sigma}$ is a self-adjoint operator on $\mathcal{H}$, we have the Mercer's decomposition (Reed & Simon, 1980) of $\widehat{\Sigma} = \sum_{i \in \widehat{N}} \widetilde{\lambda}_i \left\langle \cdot, \widehat{\phi}_i \right\rangle_{\mathcal{H}} \widehat{\phi}_i$, where $\widehat{N}$ is an at most countable set, $\{\widetilde{\lambda}_i, \ i \in \widehat{N}\}$ is non-increasing and summable, $\{\widehat{\phi}_i, \ i \in \widehat{N}\}$ are the corresponding orthonormal eigenfunctions. Then, the empirical kernel PCA uses the space spanned by the first $\ell$ eigenvectors $\widehat{\phi}_1, \cdots, \widehat{\phi}_\ell$ to approximate $\text{span}\{\phi_1, \cdots, \phi_\ell\}$.

The following proposition describes the spectrum of $\widehat{\Sigma}$. Similar results can be found in page 6 of Shawe-Taylor et al. (2005).

**Proposition 2.4.** *(Shawe-Taylor et al., 2005) Let $\widehat{\lambda}_i$'s and $v_i$'s be the eigenvalues and corresponding eigenvectors of $k(\mathbf{X}, \mathbf{X})/n := (k(X_i, X_j))_{ij}/n$. Then, we have $\widetilde{\lambda}_i = \widehat{\lambda}_i$ and $\widehat{\phi}_i = v_i^{\mathsf{T}}(k(X_1, \cdot), \ldots, k(X_n, \cdot))^{\mathsf{T}}$ for any $i \leq n$; and $\widetilde{\lambda}_i = 0$ for any $i > n$.*

From Proposition 2.4, we have

$$\widehat{\Sigma} = \sum_{i=1}^{n} \widehat{\lambda}_i \left\langle \cdot, \widehat{\phi}_i \right\rangle_{\mathcal{H}} \widehat{\phi}_i, \tag{5}$$

where $\widehat{\lambda}_i$'s are the eigenvalues of $k(\mathbf{X}, \mathbf{X})/n$.

## 2.4 RECONSTRUCTION ERROR WITH THE INTERPOLATION SPACE NORM

To measure the performance of kernel PCA, we introduce the reconstruction error with the interpolation space norm. We shall first introduce the interpolation space.

The interpolation space $[\mathcal{H}]^s$ with source condition $s \geq 0$ is a natural generalization of the RKHS $\mathcal{H}$ (see, e.g., Steinwart et al. (2009); Dieuleveut et al. (2017); Dicker et al. (2017); Pillaud-Vivien et al. (2018); Lin et al. (2020); Fischer & Steinwart (2020); Celisse & Wahl (2021)). Also, some results in the approximation theory consider the $L_2(P)$ norm (which is a special case of the interpolation space norm as is shown below) when considering kernel methods (see e.g., Santin & Schaback (2016); Steinwart (2017)). For any $s \geq 0$, $[\mathcal{H}]^s$ can be defined as $[\mathcal{H}]^s := \left\{ \sum_{i \in N} \lambda_i^{(s-1)/2} a_i \phi_i \mid \sum_{i \in N} a_i^2 < \infty \right\}$, with the inner product deduced from $\langle \lambda_i^{(s-1)/2} \phi_i, \lambda_j^{(s-1)/2} \phi_j \rangle_{[\mathcal{H}]^s} := \delta_{ij}$.

It is easy to show that $[\mathcal{H}]^s$ is also a separable Hilbert space. Moreover, if we assume $s = 1$ or $s = 0$, the interpolation space norm $\|\cdot\|_{[\mathcal{H}]^s}$ will be reduced to $\|\cdot\|_{\mathcal{H}}$ and $\|\cdot\|_{L_2(P)}$ respectively.

Now we are prepared to define the reconstruction error of kernel PCA under the interpolation space norm. Let $B_\ell := \{(\psi_1, \ldots, \psi_\ell) \mid (\psi_1, \ldots, \psi_\ell) \text{ is an orthonormal basis of a } \ell\text{-dimension subspace of } \mathcal{H}\}$. For any $(\psi_1, \ldots, \psi_\ell) \in B_\ell$, define the reconstruction error as

$$\mathcal{R}_s(\psi_1, \ldots, \psi_\ell) := \mathbb{E}_{X \sim P} \|k(\cdot, X) - \Pi(\psi_1, \ldots, \psi_\ell) k(\cdot, X)\|^2_{[\mathcal{H}]^s},$$

where $\Pi(\psi_1, \ldots, \psi_\ell) := \sum_{i=1}^{\ell} \langle \cdot, \psi_i \rangle_{\mathcal{H}} \psi_i$.

The following theorem shows that the largest $\ell$ eigenfunctions of $\Sigma$ minimize the reconstruction error.

**Theorem 2.5.** *For any $0 \leq s \leq 1$, we have $\mathcal{R}_s(\phi_1, \ldots, \phi_\ell) = \min_{B_\ell} \mathcal{R}_s(\psi_1, \ldots, \psi_\ell)$.*

When $s = 1$, the reconstruction error of the kernel PCA can be rewritten as

$$\mathcal{R}(\psi_1, \ldots, \psi_\ell) := \mathbb{E}_{X \sim P} \|k(\cdot, X) - \Pi(\psi_1, \ldots, \psi_\ell) k(\cdot, X)\|^2_{\mathcal{H}},$$

which is the same as the one given in Reiss & Wahl (2020). A similar result as Theorem 2.5 under such setting is attained by applying the method of Lagrange multipliers, which can hardly be generalized to the interpolation space norm case. Hence, it calls for a new method for the reconstruction error under the interpolation space norm. We defer the rigorous proof of Theorem 2.5 to Appendix A.1.

*Remark* 2.6. We notice that Sriperumbudur & Sterge (2022) first claimed the same results as Theorem 2.5 when $s = 0$, and further claimed that their proof could be extended to arbitrary $0 \leq s \leq 1$. However, we notice that their proof possesses some gaps ( the gap is mainly due to the wrong decomposition of some operators and sets, see Appendix C.1 for details).

# 3  MAIN RESULTS

The main goal of this paper is to derive an upper bound of the reconstruction error of kernel PCA and present two interesting applications of it.

## 3.1  RECONSTRUCTION ERROR OF THE EMPIRICAL KERNEL PCA

We begin by the following result, which gives the lower and upper bound on the empirical error. Its proof is deferred to Appendix A.2.

**Proposition 3.1.** *For any $0 \leq s \leq 1$, we have the following statements:*

(i) *If we denote $R_{\Sigma,\ell,s} = \mathcal{R}_s (\phi_1, \ldots, \phi_\ell)$, then we have $R_{\Sigma,\ell,s} = \sum_{j \geq \ell+1} \lambda_j^{2-s}$.*

(ii) *Denote $R_{\widehat{\Sigma},\ell,s} = \mathcal{R}_s \left( \widehat{\phi}_1, \ldots, \widehat{\phi}_\ell \right)$, where $\widehat{\phi}_i$'s are the eigenfunctions of $\widehat{\Sigma}$ defined in (5). For any $t > 0$, denote $\mathcal{N}_\Sigma(t) = \left\| \Sigma^{\frac{1}{2}} (\Sigma + tI)^{-\frac{1}{2}} \right\|_{\mathcal{L}^2(\mathcal{H})}^2$. Suppose further that the following assumption $(C)$ holds:*

$$\text{There exists } \mathcal{C} \text{ (does not depend on } \ell\text{) such that } \widehat{\lambda}_{\ell+1} \leq \mathcal{C}\lambda_{\ell+1}. \tag{C}$$

*For any $\delta > 0$ and any $\ell$ satisfying $\frac{\max\{12\kappa^2, 8\kappa/\log n\}}{n} \log \frac{n}{\delta} \leq \lambda_{\ell+1}$, we have*

$$R_{\widehat{\Sigma},\ell,s} \leq 4 \left( \mathcal{C} + 1 \right)^2 \mathcal{N}_\Sigma(\lambda_{\ell+1}) \cdot \lambda_{\ell+1}^{2-s},$$

*with probability at least $1 - \delta$.*

Proposition 3.1 provides upper and lower bounds of the reconstruction error of empirical kernel PCA with $[\mathcal{H}]^s$-norm. Noticing that $\mathcal{N}_\Sigma(t)$ can be upper bounded by $\mathcal{N}_\Sigma(t) = \sum_{i \in N} \frac{\lambda_i}{t+\lambda_i} \leq \sum_{i \in N} \frac{\lambda_i}{t} \leq \frac{\kappa}{t}$, we can attain the bound of $\lambda_{\ell+1}^{1-s}$. When more information about the eigenvalues are given, we might have a tighter upper bound of $\mathcal{N}_\Sigma(t)$.

**The Necessity of Assumption $(C)$**  We notice that Thm 6.(ii) in Sriperumbudur & Sterge (2022), Thm 8.(ii) in Sterge & Sriperumbudur (2022), and Thm 3.1 in Rudi et al. (2013) claimed similar results as Proposition 3.1 by arguing that the condition $\widehat{\lambda}_{\ell+1} \leq \mathcal{C}\lambda_{\ell+1}$ holds with high probability.

- However, their proof for the above condition, mostly based on Lemma 3.5 in Rudi et al. (2013), exists gaps (see Appendix C.2 for details). (The gap is mainly due to the wrong claim that a specific operator is positive semi-definite.)
- Hence, we explicitly exhibit assumption $(C)$ to stress its necessity.

*Remark* 3.2. Sriperumbudur & Sterge (2022) proposed to use the U-statistics $\widehat{\Sigma}^{\text{center}} := \frac{1}{2n(n-1)} \sum_{i \neq j}^n (\Phi(X_i) - \Phi(X_j)) \otimes_{\mathcal{H}} (\Phi(X_i) - \Phi(X_j))$ rather than the empirical version in our case. However, due to the great difficulty of estimating the eigenvalues of $\widehat{\Sigma}^{\text{center}}$, assumption $(C)$ is hard to be verified. Such difficulties were wrongly skipped by Sriperumbudur & Sterge (2022) since they took assumption $(C)$ for granted. In order to conquer such difficulties, we use the (non-centralized) empirical operator $\widehat{\Sigma}$ to serve as the empirical covariance operator. The eigenvalues of $\widehat{\Sigma}$ and $\Sigma$ can be derived from the kernel and the empirical kernel, making it possible for us to verify assumption $(C)$ and to derive an upper bound for the reconstruction error in different cases.

**Several Important Settings under which Assumption $(C)$ Holds**  In the following two subsections, we will present two applications of Proposition 3.1:

(i) The first application is more classical, i.e., we consider the situation where the eigenvalues of kernel is polynomially decaying.

(ii) The second application is more interested, i.e., we consider the reconstruction error of empirical kernel PCA for large dimensional data where sample size $n \asymp d^\gamma$ for some $\gamma > 1$.

## 3.2 KERNEL PCA UNDER POLYNOMIAL EIGENVALUE DECAY ASSUMPTION

In the classical fixed-dimensional setting where the dimension $d$ of the data is fixed, one of the typical assumptions on the kernel function is the following polynomial eigendecay assumption (Caponnetto & De Vito, 2007; Fischer & Steinwart, 2020; Zhang et al., 2023).

**Assumption 3.3** (Polynomial eigendecay assumption). There is some $\beta > 1$ and constants $c_\beta, C_\beta > 0$ such that $c_\beta j^{-\beta} \leq \lambda_j \leq C_\beta j^{-\beta}$, $j = 1, \cdots$, where $\lambda_j$ is the eigenvalue of $\Sigma$ defined in (4).

Such a polynomial decay is satisfied for the well-known Sobolev kernel with smoothness $r > d/2$ (we have $\beta = 2r/d$, see, e.g., Edmunds & Triebel (1996); Fischer & Steinwart (2020)), Laplace kernel, and, of most interest, neural tangent kernels for fully-connected multilayer neural networks (we have $\beta = (d+1)/d$, see, e.g., Bietti & Mairal (2019); Bietti & Bach (2020); Lai et al. (2023)).

With Assumption 3.3, we can calculate the quantities $\sum_{j \geq \ell+1} \lambda_j^2$ and $\mathcal{N}_\Sigma(\lambda_{\ell+1})$ in Proposition 3.1 explicitly. In particular, we can further show the optimality of the empirical kernel PCA with the polynomial eigenvalue decay assumptions (here the optimality is referred to the one introduced in Sriperumbudur & Sterge (2022)). The proof of following corollary can be found in Appendix A.3.

**Corollary 3.4.** *Suppose the eigenvalues of $\Sigma$ satisfy Assumption 3.3. We have:*

- *For any $\tau > 0$, if $n \geq \mathcal{C}_3 \ell^{2\beta}$ ($\mathcal{C}_3$ is a constant depending on $\beta$, $c_\beta$, $C_\beta$, $\tau$ and $\kappa$) , we have $\widehat{\lambda}_{\ell+1} \leq 2\lambda_{\ell+1}$ holds with probability $1 - 2e^{-\tau}$.*

- *For any $\delta > 0$, there exist constants $\mathcal{C}_{small}$, and $\mathcal{C}_{large}$ only depending on $\beta$, $c_\beta$, $C_\beta$, $\tau$, $\kappa$ and $\delta$, such that for all $n$ satisfying $n \geq \mathcal{C}_3 \ell^{2\beta}$, we have*

$$\mathcal{C}_{small} \ell^{-(2-s)\beta+1} \leq R_{\Sigma,\ell,s} \leq R_{\widehat{\Sigma},\ell,s} \leq \mathcal{C}_{large} \ell^{-(2-s)\beta+1},$$

*with probability at least $1 - \delta - 2e^{-\tau}$.*

The first statement in Corollary 3.4 ensures us that we can apply the Proposition 3.1. The second statement in Corollary 3.4 shows that when $n \succeq \ell^{2\beta}$, the convergence rate (in terms of $\ell$) of $R_{\widehat{\Sigma},\ell}$ is the same as the convergence rate of the optimal quantity $R_{\Sigma,\ell}$.

*Remark 3.5.* When $s = 1$, we can attain the bound of $\ell^{-\beta+1}$, which is in accordance with the bound in Reiss & Wahl (2020) under Assumption 3.3 (see Proposition 2.2 and Remark 2.3).

## 3.3 KERNEL PCA IN THE LARGE DIMENSIONAL SETTING

We consider the reconstruction error of the kernel PCA in large dimensional setting where $n \asymp d^\gamma$ for some $\gamma > 1$. Let us work with an inner product kernel $k^{\text{in}} : \mathbb{S}^d \times \mathbb{S}^d \to \mathbf{R}$ satisfying $k^{\text{in}}(x,y) = \Psi(\langle x, y \rangle)$, where $\Psi : [-1,1] \to \mathbf{R}$. We denote the decomposition of $k^{\text{in}}$ as $k^{\text{in}}(x,y) = \sum_{k=0}^\infty \mu_k \sum_{j=1}^{N(d,k)} Y_{k,j}(x) Y_{k,j}(y)$, where $Y_{k,j}$ for $j = 1, \cdots, N(d,k)$ are spherical harmonic polynomials of degree $k$ and $\mu_k$'s are the eigenvalues of $k$ with multiplicity $N(d,0) = 1$; $N(d,k) = \frac{2k+d-1}{k} \cdot \frac{(k+d-2)!}{(d-1)!(k-1)!}, k = 1, 2, \cdots$.

*Remark 3.6.* We consider the inner product kernels on the sphere mainly because the harmonic analysis is clear on the sphere (e.g., properties of spherical harmonic polynomials are more concise than the orthogonal series on general domains). This makes Mercer's decomposition of the inner product more explicit rather than several abstract assumptions (e.g., Mei & Montanari (2022)). We also notice that very few results are available for Mercer's decomposition of a kernel defined on the general domain, especially when the dimension of the domain is taking into consideration. e.g., even the eigen-decay rate of the neural tangent kernels is only determined for the spheres. Restricted by this technical reason, most works analyzing the kernel method in large dimensional settings focus on the inner product kernels on spheres (Liang et al., 2020; Ghorbani et al., 2021; Misiakiewicz, 2022;

Xiao et al., 2022; Lu et al., 2023, etc.). Though there might be several works that tried to relax the spherical assumption (e.g., Liang et al. (2020); Aerni et al. (2022); Barzilai & Shamir (2023)), we can find that most of them hide the essential requirements in the assumptions.

To avoid unnecessary notation, we introduce the following assumption on the inner kernel $k^{\mathrm{in}}$:

**Assumption 3.7.** Coefficients $\{a_i, \; i = 0, 1 \ldots\}$ in Taylor expansion $\Psi(t) = \sum_{i=0}^{\infty} a_i t^i$ are positive.

The purpose of Assumption 3.7 is to keep the main results and proofs clean. Notice that, by Theorem 1.b in Gneiting (2013), the inner product kernel $K$ on the sphere is positive semidefinite for all dimensions if and only if all coefficients $\{a_j, j = 0, 1, 2, ...\}$ are non-negative. One can easily extend our results in this paper when certain coefficients $a_k$'s are zero (e.g., one can consider the two-layer NTK defined as in Section 5 of Lu et al. (2023), with $a_i = 0$ for any $i = 3, 5, 7, \cdots$).

Now, we are prepared to give one of the main results of this paper.

**Theorem 3.8.** *If $0 \leq s \leq 1$, consider the kernel defined on the sphere $\mathbb{S}^{d-1}$. Suppose $n \asymp d^\gamma$. For any $\ell$, let $q$ be an integer satisfying $N(q) \leq l < N(q+1)$, where $N(q) = \sum_{k=0}^{q} N(d, k)$. If we have $q \leq \lfloor \frac{\gamma}{2} \rfloor$ and $N(q+1) - \ell \asymp d^{q+1}$, then the following statements hold:*

*(i) $R_{\Sigma, \ell, s} \asymp d^{-(q+1)(1-s)}$.*

*(ii) For any $\delta > 0$, there exist a constant $C$ only depending on $c_1, c_2, \gamma$ and $\delta$, and a constant $C_1$ only depending on $c_1, c_2, \gamma$, such that for any $d \geq C$, we have*

$$R_{\widehat{\Sigma}, \ell, s} \asymp d^{-(q+1)(1-s)},$$

*with probability at least $1 - \delta - C_1 d^\gamma e^{-d^{\gamma - q - 1}}$.*

Theorem 3.8 provides a tight convergence rate on the reconstruction error of large dimensional (empirical) kernel PCA. Notice that when $s = 1$, the reconstruction error $R_{\Sigma, \ell, s} \asymp R_{\widehat{\Sigma}, \ell, s} = \Theta(1)$, which implies that adopting the $\mathcal{H}$-norm leads to inconsistent reconstruction error when considering kernel PCA in large dimensions.

*Remark* 3.9. We can still derive the same upper bound of empirical error when the condition $N(q+1) - \ell \asymp d^{q+1}$ is not satisfied. However, under such setting, the optimal error has a better performance, whose convergence rate ranges from $d^{-(q+1)(1-s)}$ to $d^{-(q+2)(1-s)}$. The reason of this phenomenon is that the $N(q+1) - \ell$ tail eigenfunctions of $\mu_{q+1}$ have a far greater impact on empirical kernel PCA rather than kernel PCA. However, notice that $N(q+1) - N(q) \asymp d^{q+1}$, we find that $N(q+1) - \ell \asymp d^{q+1}$ holds true for large portion of $\ell$ satisfying that $N(q) \leq \ell \leq N(q+1)$.

As is shown in Figure 1(c), a periodic plateau phenomenon under large dimensional kernel PCA setting can be observed: the rate of the reconstruction error remains unchanged over certain intervals of $\ell$. A similar periodic plateau phenomenon was reported by Lu et al. (2023); Zhang et al. (2024a) when considering large-dimensional spectral algorithms: the rate of the excess risk remains unchanged over certain intervals of $\gamma$. The similarity is due to the following reasons. When a faster rate is required, $\ell$ must increase so that $q$ becomes larger. However, due to the inequality $q \leq \lfloor \gamma/2 \rfloor$, larger q requires increasing $\gamma$ above a certain threshold. Hence, the rate of reconstruction error remains unchanged over certain intervals of $\gamma$.

These behaviors on the kernel PCA under large dimension setting indicate that to improve the reconstruction error rate, it is necessary to increase $\gamma$ (or equivalently, the sample size $n$) beyond a specific threshold. Also, we believe that the periodic plateau behavior is widely exhibited in large dimensional kernel-related algorithms.

# 4 NUMERICAL EXPERIMENT

In this section, we provide a brief numerical experiment to verify the results in Theorem 3.8.

We assume that each $x_i$ is i.i.d. sampled from the uniform distribution on $\mathbb{S}^d$. We consider the following two inner product kernels:

- The RBF kernel with a fixed bandwidth: $k^{\mathrm{rbf}}(x, y) = \exp\left(-\|x - y\|_2^2 / 2\right), \; x, y \in \mathbb{S}^d$.

- The three-layer neural tangent kernel (NTK) $k^{\mathrm{ntk}}$ defined in Bietti & Bach (2020).

It can be verified that both of the above kernels satisfy Assumption 3.7 (see, e.g., Zhang et al. (2024b); Bietti & Bach (2020)). We let $n = d^\gamma$ with $\gamma = 2.1, 1.5$, and we choose the dimension $d$ from 10 to 60 with step 1, from 50 to 100 with step 5, respectively. We set $s = 0$ and $\ell = d^\xi$ with $\xi = 0.4, 1.2$. Notice that we only consider $\xi = 0.4$ when $\gamma = 1.5$ since $q$ in Theorem 3.8 should satisfy $q \le \lfloor \frac{\gamma}{2} \rfloor$.

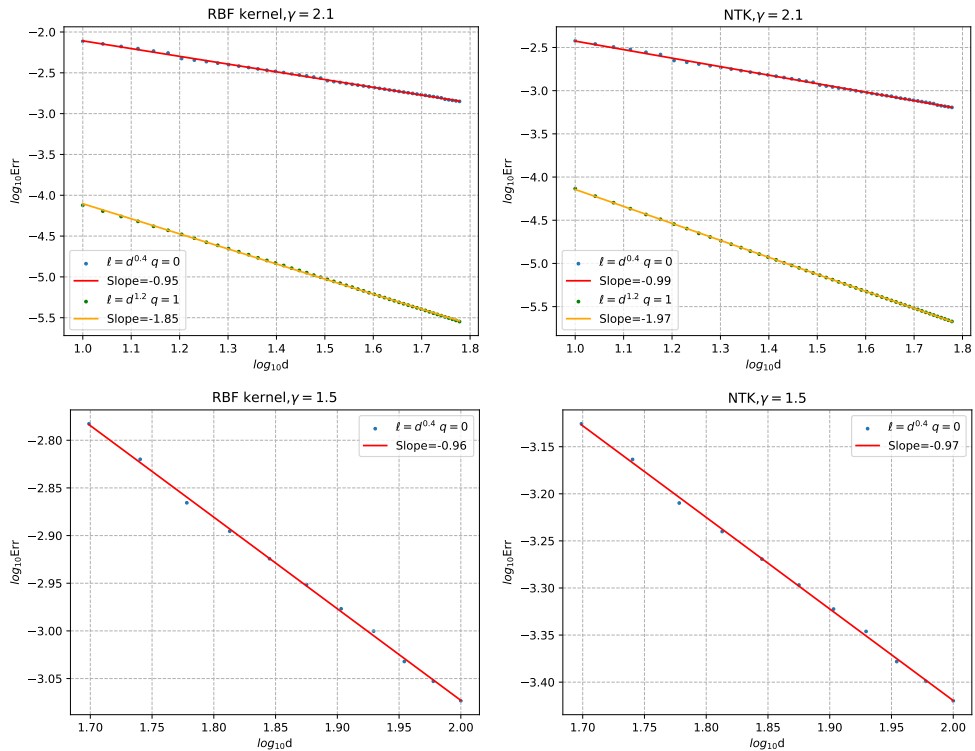

Figure 2: The reconstruction error of kernel PCA for different kernels and different $\gamma$. The first and second rows correspond to $\gamma = 2.1$ and $\gamma = 1.5$; while the first and the second columns use the RBF kernel and NTK, respectively. Each point represents the mean of 10 i.i.d. experiments. We perform logarithmic least-square $\log_{10} \mathrm{Err} = r \log_{10} d + b$ to fit the generalization error with respect to the dimension, thus the slope $r$ will be the convergence rate of reconstruction error with respect to $d$.

Figure 2 displays the results. It can be concluded that the convergence rates of the reconstruction error in all cases are close to the theoretical convergence rate $-(q+1)$ in Theorem 3.8.

## 5 CONCLUSION

Reconstruction errors of PCA and kernel PCA have become an active research topic recently. Comparing with the studies in the PCA, few results have been obtained in the reconstruction errors of kernel PCA. In this paper, we provided both lower and upper bound of the reconstruction error of empirical kernel PCA. Furthermore, we utilize it to analyze two prevalent situations: 1. when the dimension is fixed, the eigenvalues of the kernel is polynomially decaying; 2. when the large dimensional data is supported on the sphere $\mathbb{S}^{d-1}$. In both case, we illustrated that the bounds provide here are optimal in the sense introduced in Sriperumbudur & Sterge (2022).

There might be a few interesting questions for future research: $i)$ We considered the empirical kernel PCA in large dimensional settings, however, the performance of variants of kernel PCA, such as random feature kernel PCA and Nyström kernel PCA, remains unknown in the large dimension settings. The analysis of such variants may give a closer look on how the kernel PCA method acts in large dimension cases. $ii)$ It would be of great interest to derive some minimax optimal results of the reconstruction error of empirical PCA and empirical kernel PCA. To the best of our knowledge, even the minimax optimality of the empirical PCA has only been showed for the spiked models.

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

## A  PROOFS OF MAIN RESULTS

### A.1  PROOF OF THEOREM 2.5

We first introduce some notations. Denote $T : L^2(\mathcal{X}, P) \to L^2(\mathcal{X}, P)$, $(Tf)(x) = \int_{\mathcal{X}} k(x, y) f(y) dP(y)$. $\mathcal{J} : \mathcal{H} \to L^2(\mathcal{X}, P)$ be the inclusion map. It can be shown that $T = \mathcal{J} \mathcal{J}^*$, and hence that $T$ is self-adjoint, positive, trace-class and compact, see Steinwart & Scovel (2012). The Mercer's decomposition asserts that

$$k(x, y) = \sum_{i \in N} \lambda_i e_i(x) e_i(y)$$

$$T = \sum_{i \in N} \lambda_i \langle \cdot, e_i \rangle_{L^2(\mathcal{X}, P)} e_i.$$

where $N$ is an at most countable set, $\{e_i, \ i \in N\}$ are the corresponding orthonormal eigenfunctions under the space of $L^2(\mathcal{X}, P)$. Then, it is well known that $\forall i \in N, \phi_i = \sqrt{\lambda_i} e_i$.

Now let's begin to prove Theorem 2.5.

*Proof of Theorem 2.5.* Let $\psi_j = \sum_q a_{qj} \sqrt{\lambda_q} e_q$ be an orthonormal basis in $\mathcal{H}$. Define $A_1$ as $(a_{qj})_{j \leq \ell}$, $A_2$ as $(a_{qj})_{j \geq \ell+1}$, A as $(a_{qj})$. We have

$$k(\cdot, x) - \Pi(\psi_1, \cdots, \psi_\ell) k(\cdot, x) = \sum_{j > \ell} \psi_j(x) \psi_j = \sum_{j > \ell} \psi_j(x) \sum_{q \in N} a_{qj} \sqrt{\lambda_q} e_q$$

$$= \sum_{q \in N} \left( \sum_{j > \ell} \psi_j(x) a_{qj} \right) \sqrt{\lambda_q} e_q.$$

Hence, we have

$$\|k(\cdot, x) - \Pi(\psi_1, \cdots, \psi_\ell) k(\cdot, x)\|_{[\mathcal{H}]^s}^2 = \sum_{q=1} \lambda_q^{1-s} \left( \sum_{j > \ell} \psi_j(x) a_{qj} \right)^2$$

$$= \sum_{q=1} \lambda_q^{1-s} \left( \sum_{j > \ell} \sum_{p=1} a_{pj} \sqrt{\lambda_p} e_p(x) a_{qj} \right)^2$$

$$= \sum_{q=1} \lambda_q^{1-s} \left( \sum_{p=1} \sum_{j > \ell} a_{pj} a_{qj} \sqrt{\lambda_p} e_p(x) \right)^2$$

Notice that $A$ is orthogonal, hence $\sum_{j=1} a_{pj} a_{qj} = 1$, and the reconstruction error is

$$\mathcal{R}_s(\psi_1, \ldots, \psi_\ell) = \sum_{q,p} \left( \delta_{pq} - \sum_{j \leq \ell} a_{pj} a_{qj} \right)^2 \lambda_p \lambda_q^{1-s} = tr((A_1 A_1^{\mathrm{T}} - Id) \Lambda (A_1 A_1^{\mathrm{T}} - Id) \Lambda^{1-s}),$$

where $\Lambda$ is the diagonalized operator of the eigenvalues.

$$\frac{\partial \mathcal{R}_s\left(\psi_1, \ldots, \psi_\ell\right)}{\partial a_{u,j}} = \frac{\partial}{\partial a_{uj}} \sum_u \sum_v \left(\delta_{uv} - \sum_h a_{uh} a_{vh}\right)^2 \lambda_u \lambda_v^{1-s}$$

$$= 4\lambda_u^{2-s} a_{uj} \left(\sum_{h=1}^\ell a_{uh}^2 - 1\right)^2 + 2\lambda_u \sum_{v \neq u} \lambda_v^{1-s} \left(\sum_{h=1}^\ell a_{uh} a_{vh}\right) a_{vj}$$

$$+ 2\lambda_u^{1-s} \sum_v \lambda_v \left(\sum_{h=1}^\ell a_{uh} a_{vh}\right) a_{vj}$$

$$= 2\lambda_u \sum_{v \in N} \lambda_v^{1-s} \left(\sum_{h=1}^\ell a_{uh} a_{vh} - \delta_{uv}\right) a_{vj} + 2\lambda_u^{1-s} \sum_{v \in N} \lambda_v \left(\sum_{h=1}^\ell a_{uh} a_{vh} - \delta_{uv}\right) a_{vj}$$

$$= 2 \sum_{v \in N} \lambda_u \lambda_v \left(\lambda_u^{-s} + \lambda_v^{-s}\right) \left(\sum_{h=1}^\ell a_{uh} a_{vh} - \delta_{uv}\right) a_{vj}.$$

Hence $\nabla_{A_1} \mathcal{R}_s\left(\psi_1, \ldots, \psi_\ell\right) = 2\Lambda H \Lambda A_1$, where $H = \Lambda^{-s}(A_1 A_1^{\mathsf{T}} - Id) + (A_1 A_1^{\mathsf{T}} - Id)\Lambda^{-s}$ and $H_{uv} = (\lambda_u^{-s} + \lambda_v^{-s})\left(\sum_{h=1}^\ell a_{uh} a_{vh} - \delta_{uv}\right)$.

Now consider the Lagrange multipliers of the optimization problem. Suppose that $\mu = (\mu_{ij})$,

$$\mathcal{L} = \sum_{i=1}^\ell \sum_{j=1}^\ell \mu_{ij} \left(\sum_{p \in N} a_{pi} a_{pj} - \delta_{ij}\right) = \sum_{p \in N} \sum_{i,j} a_{pi} a_{pj} \mu_{ij} - \sum_{i=1}^\ell \mu_{ii}.$$

We have $\frac{\partial \mathcal{L}}{\partial a_{uj}} = 2 \sum_{i=1}^\ell a_{ui} \mu_{ij}$, $\nabla_{A_1} \mathcal{L} = 2A_1 \mu$.

By the Lagrange multipliers, we have

$$\Lambda(\Lambda^{-s}((A_1 A_1^{\mathsf{T}} - Id) + (A_1 A_1^{\mathsf{T}} - Id))\Lambda^{-s} A_1)\Lambda = -A_1 \mu. \tag{6}$$

When $s = 1$, we have $(A_1 A_1^{\mathsf{T}} - Id)\Lambda A_1 + A_1 \mu = 0$. Multiplying $A_1^{\mathsf{T}}$ and we get $\mu = 0$, and hence $(A_1 A_1^{\mathsf{T}} - Id)\Lambda A_1 = 0$.

When $s = 0$, we have $\Lambda(A_1 A_1^{\mathsf{T}} - Id)\Lambda A_1 + A_1 \mu = 0$. Hence, $((A_1 A_1^{\mathsf{T}} - Id)\Lambda A_1)^{\mathsf{T}}\Lambda(A_1 A_1^{\mathsf{T}} - Id)\Lambda A_1 = 0$. We get $\Lambda^{1/2}(A_1 A_1^{\mathsf{T}} - Id)\Lambda A_1 = 0$, which leads to $(A_1 A_1^{\mathsf{T}} - Id)\Lambda A_1 = 0$.

Once we have $(A_1 A_1^{\mathsf{T}} - Id)\Lambda A_1 = 0$, the reconstruction error satisfies

$$\mathcal{R}_s\left(\psi_1, \ldots, \psi_\ell\right) = tr((A_1 A_1^{\mathsf{T}} - Id)\Lambda(A_1 A_1^{\mathsf{T}} - Id)\Lambda^{1-s}) = tr(-(A_1 A_1^{\mathsf{T}} - Id)\Lambda^{2-s}) \geq \sum_{i > \ell} \lambda_i^{2-s}.$$

For general $0 < s < 1$, the space spanned by $A_1$ is an invariant subspace of operator $\Lambda(\Lambda^{-s}(A_1 A_1^{\mathsf{T}} - Id) + (A_1 A_1^{\mathsf{T}} - Id)\Lambda^{-s})\Lambda$. Hence, the space spanned by $A_2$ is also an invariant subspace of $\Lambda(\Lambda^{-s}(A_1 A_1^{\mathsf{T}} - Id) + (A_1 A_1^{\mathsf{T}} - Id)\Lambda^{-s})\Lambda$. Hence, we have the following equation

$$\Lambda^{1-s}(A_1 A_1^{\mathsf{T}} - Id)\Lambda + \Lambda(A_1 A_1^{\mathsf{T}} - Id)\Lambda^{1-s} = -A_1 \mu A_1^{\mathsf{T}} - A_2 \widetilde{\mu} A_2^{\mathsf{T}}$$

$$= -(A_1, A_2) \begin{pmatrix} \mu & 0 \\ 0 & \widetilde{\mu} \end{pmatrix} \begin{pmatrix} A_1^{\mathsf{T}} \\ A_2^{\mathsf{T}} \end{pmatrix}. \tag{7}$$

Notice that minimizing $\mathcal{R}_s(\psi_1, \cdots, \psi_\ell)$ is equivalent to maximizing $\mathcal{R}_s(\psi_{\ell+1}, \cdots)$, hence by considering the Lagrange multipliers method of $A_2$, we derive a similar equation

$$\Lambda^{1-s}(A_2 A_2^{\mathsf{T}} - Id)\Lambda + \Lambda(A_2 A_2^{\mathsf{T}} - Id)\Lambda^{1-s} = -A_2 \nu A_2^{\mathsf{T}} - A_1 \widetilde{\nu} A_1^{\mathsf{T}}$$

$$= -(A_1, A_2) \begin{pmatrix} \widetilde{\nu} & 0 \\ 0 & \nu \end{pmatrix} \begin{pmatrix} A_1^{\mathsf{T}} \\ A_2^{\mathsf{T}} \end{pmatrix}. \tag{8}$$

Adding (7) and (8), we get

$$2\Lambda^{2-s} = (A_1, A_2) \begin{pmatrix} \mu + \widetilde{\nu} & 0 \\ 0 & \nu + \widetilde{\mu} \end{pmatrix} \begin{pmatrix} A_1^{\mathrm{T}} \\ A_2^{\mathrm{T}} \end{pmatrix},$$

which implies that $A$ is block-diagonal, and thus $A_1$ is non-zero only on $\ell$ $e_q$'s. Hence, we have $\mathcal{R}_s(\psi_1, \ldots, \psi_\ell) \geq \sum_{i>\ell} \lambda_i^{2-s}$.

Also, choose $a_{ij} = \delta_{ij}$, we find that the lower bound can be attained.

□

## A.2 PROOF OF PROPOSITION 3.1

From Lemma A.1 we have

$$R_{\widehat{\Sigma}, \ell, s} = \left\| \Sigma^{\frac{1}{2}} \left( I - \Pi\left(\widehat{\phi}_1, \ldots, \widehat{\phi}_\ell\right)\right) \Sigma^{\frac{1-s}{2}} \right\|_{\mathcal{L}^2(\mathcal{H})}^2, \qquad (9)$$

and the right-hand side of (9) can be further bounded by the following three terms with any $t > 0$:

$$\mathbf{I} = \left\| \Sigma^{\frac{1}{2}} (\Sigma + tI)^{-\frac{1}{2}} \right\|_{\mathcal{L}^2(\mathcal{H})}^2 = \mathcal{N}_\Sigma(t)$$

$$\mathbf{II} = \left\| (\Sigma + tI)^{\frac{1}{2}} \left( I - \Pi\left(\widehat{\phi}_1, \ldots, \widehat{\phi}_\ell\right)\right) (\Sigma + tI)^{\frac{1}{2}} \right\|_{\mathcal{L}^\infty(\mathcal{H})}^2$$

$$\mathbf{III} = \left\| (\Sigma + tI)^{-\frac{1}{2}} \Sigma^{\frac{1-s}{2}} \right\|_{\mathcal{L}^\infty(\mathcal{H})}^2.$$

Notice that we have

$$\mathbf{III} = \sup_{i \in N} \frac{\lambda_i^{1-s}}{\lambda_i + t} = \sup_{i \in N} \frac{\lambda_i^{1-s}}{(\lambda_i + t)^{1-s}} \frac{1}{(\lambda_i + t)^s} \leq \frac{1}{t^s}.$$

For any $\delta > 0$, when $\frac{\max\{12\kappa^2, 8\kappa/\log n\}}{n} \log \frac{n}{\delta} \leq t \leq \|\Sigma\|_\infty$, we have

$$\mathbf{II} \leq \left\| (\Sigma + tI)^{\frac{1}{2}} (\widehat{\Sigma} + tI)^{-\frac{1}{2}} \right\|_{\mathcal{L}^\infty(\mathcal{H})}^4 \left\| (\widehat{\Sigma} + tI)^{\frac{1}{2}} \left( I - \Pi\left(\widehat{\phi}_1, \ldots, \widehat{\phi}_\ell\right)\right) (\widehat{\Sigma} + tI)^{\frac{1}{2}} \right\|_{\mathcal{L}^\infty(\mathcal{H})}^2$$

$$\leq \left\| (\Sigma + tI)^{\frac{1}{2}} (\widehat{\Sigma} + tI)^{-\frac{1}{2}} \right\|_{\mathcal{L}^\infty(\mathcal{H})}^4 \left( \widehat{\lambda}_{\ell+1} + t \right)^2$$

$$\leq 4 \left( \widehat{\lambda}_{\ell+1} + t \right)^2$$

with probability at least $1 - \delta$, where the last inequality comes from Lemma A.2.

Combining all these, taking $t = \lambda_{\ell+1}$, we have

$$R_{\widehat{\Sigma}, \ell, s} \leq 4 \left(\mathcal{C} + 1\right)^2 \mathcal{N}_\Sigma(\lambda_{\ell+1}) \cdot \lambda_{\ell+1}^{2-s},$$

with probability at least $1 - \delta$.

□

### A.2.1 TECHNICAL RESULTS FOR THE PROOF OF PROPOSITION 3.1

**Lemma A.1** (Restate Proposition 11 (i) in Sriperumbudur & Sterge (2022))**.** *We have the following equation*

$$\mathcal{R}_s(\psi_1, \ldots, \psi_\ell) = \left\| \Sigma^{\frac{1}{2}} (I - \Pi(\psi_1, \ldots, \psi_\ell)) \Sigma^{\frac{1-s}{2}} \right\|_{\mathcal{L}^2(\mathcal{H})}^2.$$

For readers' convenience, we provide a proof for Lemma A.1 as follows.

*Proof of Lemma A.1.* Denote $\psi_i(x) = \sum_{j \in N} a_{ij} e_j(x) = \sum_{j \in N} \frac{a_{ij}}{\sqrt{\lambda_j}} \phi_j(x) =: \sum_{j \in N} b_{ij} \phi_j(x)$, then we have

$$
\begin{aligned}
\mathbf{RHS} &= \sum_{i \in N} \langle \Sigma^{\frac{1}{2}} \left( I - \Pi(\psi_1, \ldots, \psi_l) \right) \Sigma^{\frac{1-s}{2}} \phi_i, \Sigma^{\frac{1}{2}} \left( I - \Pi(\psi_1, \ldots, \psi_l) \right) \Sigma^{\frac{1-s}{2}} \phi_i \rangle_{\mathcal{H}} \\
&= \sum_{i \in N} \lambda_i^{1-s} \langle \Sigma^{\frac{1}{2}} \left( I - \Pi(\psi_1, \ldots, \psi_l) \right) \phi_i, \Sigma^{\frac{1}{2}} \left( I - \Pi(\psi_1, \ldots, \psi_l) \right) \phi_i \rangle_{\mathcal{H}} \\
&= \sum_{i \in N} \lambda_i^{1-s} \langle \Sigma^{\frac{1}{2}} (\phi_i - \sum_{j=1}^{\ell} \langle \phi_i, \psi_j \rangle_{\mathcal{H}} \psi_j), \Sigma^{\frac{1}{2}} (\phi_i - \sum_{j=1}^{\ell} \langle \phi_i, \psi_j \rangle_{\mathcal{H}} \psi_j) \rangle_{\mathcal{H}} \\
&= \sum_{j \in N} \lambda_j^{1-s} \langle \Sigma^{\frac{1}{2}} (\phi_j - \sum_{i=1}^{\ell} \langle \phi_j, \psi_i \rangle_{\mathcal{H}} \psi_i), \Sigma^{\frac{1}{2}} (\phi_j - \sum_{i=1}^{\ell} \langle \phi_j, \psi_i \rangle_{\mathcal{H}} \psi_i) \rangle_{\mathcal{H}} \\
&= \sum_{j \in N} \lambda_j^{1-s} \langle \Sigma^{\frac{1}{2}} (\phi_j - \sum_{i=1}^{\ell} b_{ij} \psi_i), \Sigma^{\frac{1}{2}} (\phi_j - \sum_{i=1}^{\ell} b_{ij} \psi_i) \rangle_{\mathcal{H}} \\
&= \sum_{j \in N} \lambda_j^{1-s} \langle \Sigma^{\frac{1}{2}} (\phi_j - \sum_{i=1}^{\ell} b_{ij} \sum_{k \in N} b_{ik} \phi_k(x)), \Sigma^{\frac{1}{2}} (\phi_j - \sum_{i=1}^{\ell} b_{ij} \sum_{k \in N} b_{ik} \phi_k(x)) \rangle_{\mathcal{H}} \\
&= \sum_{j \geq 1} \lambda_j^{-s} \left[ \left( \lambda_j - \sum_{i=1}^{\ell} a_{ij}^2 \right)^2 + \sum_{j \neq k} \left( \sum_{i=1}^{\ell} a_{ij} a_{ik} \right)^2 \right] \\
&= \mathbf{LHS}.
\end{aligned}
$$

$\square$

**Lemma A.2.** *Let $\Sigma$ and $\widehat{\Sigma}$ be given in (4) and (5). Then, for any $0 < \delta < 1$ and any $\frac{\max\{12\kappa^2, 8\kappa/\log n\}}{n} \log \frac{n}{\delta} \leq t \leq \|\Sigma\|_{\mathcal{L}^\infty(\mathcal{H})}$, we have*

$$
\left\| (\Sigma + tI)^{\frac{1}{2}} \left( \widehat{\Sigma} + tI \right)^{-\frac{1}{2}} \right\|_{\mathcal{L}^\infty(\mathcal{H})}^2 \leq 2,
$$

*with probability at least $1 - \delta$.*

*Remark A.3.* We notice that Lemma 3.6 in Rudi et al. (2013) claimed a similar result as Lemma A.2 when $\kappa = 1$. We provide a rigorous proof for general $\kappa > 0$ as follows.

*Proof of Lemma A.2.* By defining the operator $B := (\Sigma + tI)^{-1/2}(\Sigma - \widehat{\Sigma})(\Sigma + tI)^{-1/2}$, it is straightforward to verify the following inequalities:

$$
\begin{aligned}
\left\| (\Sigma + tI)^{\frac{1}{2}} (\widehat{\Sigma} + tI)^{-\frac{1}{2}} \right\|_{\mathcal{L}^\infty(\mathcal{H})}^2 &= \left\| (\Sigma + tI)^{\frac{1}{2}} (\widehat{\Sigma} + tI)^{-1} (\Sigma + tI)^{\frac{1}{2}} \right\|_{\mathcal{L}^\infty(\mathcal{H})} \\
&= \left\| (I - B)^{-1} \right\|_{\mathcal{L}^\infty(\mathcal{H})} \leq \left( 1 - \|B\|_{\mathcal{L}^\infty(\mathcal{H})} \right)^{-1}
\end{aligned}
$$

The last inequality follows from the fact that $(I - B)^{-1} \preceq \left( 1 - \|B\|_{\mathcal{L}^\infty(\mathcal{H})} \right)^{-1} I$ whenever $\|B\|_{\mathcal{L}^\infty(\mathcal{H})} < 1$. We shall establish a probabilistic upper bound for $\|B\|_{\mathcal{L}^\infty(\mathcal{H})}$.

To bound $\|B\|_{\mathcal{L}^\infty(\mathcal{H})}$, we employ Lemma B.1, which is included in Appendix B for completeness. In particular, we set the parameters of Lemma B.1 as follows: Let $Y := U \otimes U$, where $U := (\Sigma + tI)^{-1/2} \Phi(X)$, be a random variable, and $X \sim P$ be the random variable from which the data is sampled. Since

$$
\|Y\|_{\mathcal{L}^\infty(\mathcal{H})} \leq \left\| (\Sigma + tI)^{-1} \right\|_{\mathcal{L}^\infty(\mathcal{H})} \|\Phi(X)\|_{\mathcal{H}}^2 \leq \kappa^2/t,
$$

we let $R := \kappa^2/t$, and $T := \mathbb{E}[Y] = (\Sigma + tI)^{-1/2} \Sigma (\Sigma + tI)^{-1/2}$.

Since

$$\mathbb{E}_{X \sim P}\left[(U \otimes U - T)^2\right] = \mathbb{E}_{X \sim P}\left[\|U\|_{\mathcal{H}}^2 U \otimes U - T^2\right] \preceq \mathbb{E}_{X \sim P}\left[\|U\|_{\mathcal{H}}^2 U \otimes U\right] \preceq RT,$$

we set $S := RT$. Finally, it is $\sigma^2 = \|RT\|_{\mathcal{L}^\infty(\mathcal{H})} \le \kappa^2/t$, and $d = \|S\|_{\mathcal{L}^1(\mathcal{H})}/\|S\|_{\mathcal{L}^\infty(\mathcal{H})} \le \frac{\left(\|\Sigma\|_{\mathcal{L}^\infty(\mathcal{H})}+t\right)\|T\|_{\mathcal{L}^1(\mathcal{H})}}{\|\Sigma\|_{\mathcal{L}^\infty(\mathcal{H})}}$. With this choice of parameters, Lemma B.1 implies that, with probability $1 - \delta$, it is:

$$\|B\|_{\mathcal{L}^\infty(\mathcal{H})} \le \frac{2\kappa^2\beta}{3tn} + \sqrt{\frac{2\kappa^2\beta}{tn}} \tag{10}$$

where $\beta = \log \frac{4\left(\|\Sigma\|_{\mathcal{L}^\infty(\mathcal{H})}+t\right)\|T\|_{\mathcal{L}^1(\mathcal{H})}}{\|\Sigma\|_{\mathcal{L}^\infty(\mathcal{H})}\delta}$.

By requiring that $t \ge 12\kappa^2\beta/n \ge 4(4 + \sqrt{15})\kappa^2\beta/3n$, it can be verified that $\mathbb{P}\left[\|B\|_{\mathcal{L}^\infty(\mathcal{H})} \le 1/2\right] \ge 1 - \delta$ by simple calculation.

Next, we shall verify that the condition $t \ge \frac{\max\{12\kappa^2, 8\kappa/\log n\}}{n} \log \frac{n}{\delta}$ is sufficient to ensure $t \ge 12\kappa^2\beta/n$. Notice that $d \le 2\|T\|_{\mathcal{L}^1(\mathcal{H})} \le \frac{2\kappa}{t}$, hence $12\kappa^2\beta/n \le (12\kappa^2/n) \cdot \log \frac{8\kappa}{\delta t}$. Also, we have $nt \ge 8\kappa$, hence $(12\kappa^2/n) \cdot \log \frac{8\kappa}{\delta t} \le \frac{\max\{12\kappa^2, 8\kappa/\log n\}}{n} \log \frac{n}{\delta}$.

Finally, since $t \ge \frac{\max\{12\kappa^2, 8\kappa/\log n\}}{n} \log \frac{n}{\delta}$ implies $\mathbb{P}\left[\|B\|_{\mathcal{L}^\infty(\mathcal{H})} \le 1/2\right] \ge 1 - \delta$, then, with probability $1 - \delta$, it holds

$$\left\|(\Sigma + tI)^{\frac{1}{2}}(\widehat{\Sigma} + tI)^{-\frac{1}{2}}\right\|_{\mathcal{L}^\infty(\mathcal{H})}^2 \le \left(1 - \|B\|_{\mathcal{L}^\infty(\mathcal{H})}\right)^{-1} \le 2$$

as claimed. $\qquad\square$

## A.3 PROOF OF COROLLARY 3.4

By Lemma B.2, we have $\sup_{j\ge 1}\left|\lambda_j - \widehat{\lambda_j}\right| \le \frac{2\sqrt{2}\kappa\sqrt{\tau}}{\sqrt{n}}$ with probability at least $1 - 2e^{-\tau}$. Thus, when $n \ge \mathcal{C}_3\ell^{2\beta}$ where $\mathcal{C}_3$ is a constant depending on $c_\beta, C_\beta, \beta, \tau, \kappa$, we have $\widehat{\lambda}_{\ell+1} \le 2\lambda_{\ell+1}$ with probability at least $1 - 2e^{-\tau}$ and $\lambda_{\ell+1} \ge \frac{\max\{12\kappa^2, 8\kappa/\log n\}}{n} \log \frac{n}{\delta}$, so by Lemma B.3 and Proposition 3.1, we get

$$R_{\widehat{\Sigma},\ell,s} \le 36\mathcal{N}_\Sigma(\lambda_{\ell+1})\lambda_{\ell+1}^{2-s} \le \mathcal{C}_{large}\ell^{-(2-s)\beta+1}$$

with probability at least $1 - \delta - 2e^{-\tau}$. Also, $R_{\Sigma,\ell,s} = \sum_{i\ge\ell+1}\lambda_i^{2-s} > \mathcal{C}_{small}\ell^{-(2-s)\beta+1}$. Hence, we reach the conclusion in the corollary. $\qquad\square$

## A.4 PROOF OF THEOREM 3.8

For (i), denote $M = N(q+1) - \ell$, we have

$$R_{\Sigma,\ell,s} = \sum_{i\ge\ell+1}\lambda_i^{2-s} = (N(q+1)-\ell)\mu_{q+1}^{2-s} + \sum_{k=q+2}\mu_k^{2-s}N(d,k) = M\mu_{q+1}^{2-s} + \sum_{k=q+2}\mu_k^{2-s}N(d,k).$$

On the one hand,

$$R_{\Sigma,\ell,s} \ge M\mu_{q+1}^{2-s} \asymp d^{-(q+1)(1-s)}.$$

On the other hand,

$$\sum_{k=q+2}\mu_k^{2-s}N(d,k) \lesssim d^{-(1-s)}\mu_{q+1}^{1-s}\sum_{k=q+2}\mu_kN(d,k) \lesssim d^{-(1-s)}\mu_{q+1}^{1-s} \asymp d^{-(q+2)(1-s)}.$$

Hence, we have

$$R_{\Sigma,\ell,s} \asymp d^{-(q+1)(1-s)}$$

For (ii), since $N(q) \leq \ell < N(q+1)$, from Lemma A.5, for any $\delta > 0$, when $d \geq \mathfrak{C}$, a sufficiently large constant only depending on $c_1, c_2, \delta$, and $\gamma$, the event $E_1 = \{\widehat{\lambda}_{\ell+1} \leq \widehat{\lambda}_{N(q)+1} < 4\mu_{q+1} = 4\lambda_{\ell+1}\}$ occurs with probability at least $1 - \delta$.

Denote $E_2 = \{R_{\widehat{\Sigma},\ell,s} \leq 100\mu_{q+1}^{1-s}\}$. Since $\mu_{q+1} \asymp d^{-q-1}$, when $\delta' \gtrsim e^{-d^{\gamma-q-1}} \cdot d^\gamma$, we have $\frac{\max\{12\kappa^2, 8\kappa/\log n\}}{n}\log\frac{n}{\delta'} \leq \mu_{q+1}$. Notice that $\mathcal{N}_\Sigma(t) = \sum_{i \in N}\frac{\lambda_i}{\lambda_i+t} \leq \sum_{i \in N}\frac{\lambda_i}{t} \leq \frac{1}{t}$. From Proposition 3.1, the event $E_1 \cap E_2$ occurs with probability at least $1 - \delta - \delta'$.

Conditioning on $E_1 \cap E_2$, we have

$$d^{-(q+1)(1-s)} \lesssim R_{\Sigma,\ell,s} \leq R_{\widehat{\Sigma},\ell,s} \lesssim d^{-(q+1)(1-s)}$$

where the last inequality is because that $\mu_{q+1} \asymp d^{-q-1}$, and we get the desired results. $\qquad\square$

### A.4.1 Technical Results for the proof of of Theorem 3.8

The following two lemmas are borrowed from Lu et al. (2023), which describe the eigenvalues and the empirical ones of the inner kernel $k^{\text{in}}$.

**Lemma A.4** (Lemma B.1 and Lemma 3.3 in Lu et al. (2023)). *Suppose that $q \in \{1,2,3,\cdots\}$ and $k \in \{1,2,3,\cdots,q,q+1\}$. Suppose that Assumption 3.7 holds. There exist constants $\mathfrak{C}$, $\mathfrak{C}_1$, and $\mathfrak{C}_2$ only depending on $q$, such that for any $d \geq \mathfrak{C}$, a sufficiently large constant only depending on $q$, we have*

$$\frac{\mathfrak{C}_1}{d^k} \leq \mu_k \leq \frac{\mathfrak{C}_2}{d^k},$$

$$\mu_j \leq \frac{\mathfrak{C}_2}{\mathfrak{C}_1}d^{-1}\mu_q, \quad j = q+1, q+2, \cdots,$$

$$\mathfrak{C}_1 d^k \leq N(d,k) \leq \mathfrak{C}_2 d^k.$$

**Lemma A.5** (Lemma C.4 in Lu et al. (2023)). *Suppose that $\gamma > 1$ and define $p := \lfloor\gamma/2\rfloor$. Suppose that Assumption 3.7 holds. For any constants $0 < c_1 \leq c_2 < \infty$ and any $\delta > 0$, there exists constant $\mathfrak{C}$ only depending on $c_1, c_2, \delta$, and $\gamma$, such that for any $d \geq \mathfrak{C}$, when $c_1 d^\gamma \leq n < c_2 d^\gamma$, we have*

$$\widehat{\lambda}_{N(q)+1} < 4\mu_{q+1}, \quad q \leq p,$$

*with probability at least $1 - \delta$, where $N(q) = \sum_{k=0}^q N(d,k)$.*

*Remark A.6.* In Lemma C.4 in Lu et al. (2023), the authors only considered $\gamma \neq 2,4,6,\cdots$ and $q = p$ for their specific motivation. After checking the proofs carefully, we find that the statements in Lemma C.4 in Lu et al. (2023) holds for any $\gamma > 0$ and any $q \leq p$. Therefore, we omit the proof for Lemma A.5.

## B  Auxiliary Results

**Lemma B.1** (Concentration Inequality for Operator Norm, Tropp (2012), Theorem 7.3.1). *Let $(Y_i)_{1 \leq i \leq n} \sim Y$ be i.i.d, $Y$ taking values in the space of bounded self-adjoint operators $\mathcal{B}(\mathcal{H})$ over a separable Hilbert space $\mathcal{H}$. Define $T := \mathbb{E}[Y]$, and let there be $S \in \mathcal{L}^2(\mathcal{H})$ such that $\mathbb{E}\left[(Y-T)^2\right] \leq S$, and a finite number $R$ such that $\|Y\|_{\mathcal{L}^\infty(\mathcal{H})} \leq R$ almost everywhere. Define $d := \|S\|_{\mathcal{L}^1(\mathcal{H})}/\|S\|_{\mathcal{L}^\infty(\mathcal{H})}$ and $\sigma^2 := \|S\|_{\mathcal{L}^\infty(\mathcal{H})}$. Then, for $0 < \delta \leq d$, the following inequality holds:*

$$P\left\{\left\|\frac{1}{n}\sum_{i=1}^n Y_i - T\right\|_{\mathcal{L}^\infty(\mathcal{H})} \leq \frac{\beta R}{n} + \sqrt{\frac{3\beta\sigma^2}{n}}\right\} \geq 1 - \delta,$$

*where $\beta := \frac{2}{3}\log\frac{4d}{\delta}$.*

**Lemma B.2** (Proposition 10 in Rosasco et al. (2010)). *For eigenvalues $\{\lambda_i\}_{i \in N}$, $\{\widehat{\lambda}_i\}_{i=1}^n$, there exists extended enumerations of two sequences (adding $0$ until the two sequences have the same length, still denoted by $\{\lambda_i\}$, $\{\widehat{\lambda}_i\}$) such that*

$$\sup_{j \geq 1} \left| \lambda_j - \widehat{\lambda}_j \right| \leq \frac{2\sqrt{2}\kappa\sqrt{\tau}}{\sqrt{n}}$$

*with probability at least $1 - 2e^{-\tau}$*

**Lemma B.3** (Proposition B.3 in Li et al. (2023a)). *Under Assumption 3.3, there exist two constants $\mathcal{C}_1$ and $\mathcal{C}_2$ only depending on $\beta$, $c_\beta$, and $C_\beta$, such that we have*

$$\mathcal{C}_1 t^{-\frac{1}{\beta}} \leq \mathcal{N}_\Sigma(t) \leq \mathcal{C}_2 t^{-\frac{1}{\beta}}.$$

## C    GAPS ON THE PROOF IN PREVIOUS WORKS

In this section, we shall point out the gaps existing in the proof in Rudi et al. (2013) and Sriperumbudur & Sterge (2022).

### C.1    GAPS ON THE PROOF OF PROPOSITION 2 IN SRIPERUMBUDUR & STERGE (2022)

The gaps on the proof of Proposition 2.(i) in Sriperumbudur & Sterge (2022), mainly comes from Lemma B.1(i) in Sriperumbudur & Sterge (2022). Notice that Proposition 2.(i) is direct corollary of a Lemma B.1(i) in Sriperumbudur & Sterge (2022), hence we describe their proof process as follows, keeping the notation consistent with Sriperumbudur & Sterge (2022).

(i) For any $Q \in \mathcal{Q}_\ell = \left\{ \sum_{i=1}^\ell \tau_i \otimes_H \tau_i : (\tau_i)_{i \in [\ell]} \subset H \right\}$, Lemma B.1(i) in Sriperumbudur & Sterge (2022) aimed to give a lower bound on the loss:

$$\mathcal{R}^A_{\alpha,\delta,\theta}(Q) = \left\| A^{\delta/2} \left( I - Q A^\alpha \right) A^{\theta/2} \right\|^2_{\mathcal{L}^2(H)}, Q \in \mathcal{Q}_\ell,$$

by separating the operators $Q$ and $A$ into several parts.

(ii) Decompose $A = A_\leq + A_>$, where $A_\leq = \sum_{i=1}^\ell \lambda_i \psi_i \otimes_H \psi_i$ and $A_> = \sum_{i>\ell} \lambda_i \psi_i \otimes_H \psi_i$. The authors claimed that there existed a separation $\mathcal{A}_i, i = 1, 2, 3$ such that we have

$$(\tau_i)_{i \in \mathcal{A}_1} \subset \text{Ran}\,(A_\leq), (\tau_i)_{i \in \mathcal{A}_2} \subset \text{Ran}\,(A_>), (\tau_i)_{i \in \mathcal{A}_3} \subset \text{Ker}\,(A).$$

If this claim held, then we could decompose $Q$ as $Q_1 + Q_2 + Q_3$, where $Q_i = \sum_{i \in \mathcal{A}_i} \tau_i \otimes_\mathcal{H} \tau_i$. However, the following counterexample shows that separation $\mathcal{A}_i, i = 1, 2, 3$ may not exist. It can be shown that there exists $A$ satisfying $\text{Ker}\,(A) = \text{span}\,(0, 0, 1), \text{Ran}\,(A) = \text{span}\{(1, 0, 0), (0, 1, 0)\}$, $\text{Ran}\,(A_\leq) = \text{span}\{(1, 0, 0)\}$, $\text{Ran}\,(A_>) = \text{span}\{(0, 1, 0)\}$. However, if we let $\tau_i = (\frac{3}{5}, \frac{4}{5}, 0)$, then $i$ is not in any $\mathcal{A}_i, i = 1, 2, 3$.

(iii) The authors also claimed that there existed four sets

$$\mathcal{B} \subseteq \{1, \ldots, \ell\}, \quad \mathcal{B}^c := \{1, \ldots, \ell\} \backslash \mathcal{B}$$
$$\mathcal{C} \subseteq \{\ell + 1, \ell + 2, \ldots\}, \quad \mathcal{C}^c := \{\ell + 1, \ell + 2, \ldots\} \backslash \mathcal{C},$$

satisfying $\text{span}\left\{(\psi_i)_{i \in \mathcal{B}}\right\} = \text{span}\left\{(\tau_i)_{i \in \mathcal{A}_1}\right\}$ and $\text{span}\left\{(\psi_i)_{i \in \mathcal{C}}\right\} = \text{span}\left\{(\tau_i)_{i \in \mathcal{A}_2}\right\}$. If this claim held, then we could futher decompose $A = A_{\leq,\mathcal{B}} + A_{\leq,\mathcal{B}^c} + A_{>,\mathcal{B}} + A_{>,\mathcal{B}^c}$, where $A_{\leq,\bullet} := \sum_{i \in \bullet} \lambda_i \psi_i \otimes_H \psi_i, \bullet \in \mathcal{B}, \mathcal{B}^c$ and $A_{>,\bullet} := \sum_{i \in \bullet} \widetilde{\lambda}_i \psi_i \otimes_H \psi_i, \bullet \in \mathcal{C}, \mathcal{C}^c$. However, the following counterexample shows that the above claim may not hold. Let $\ell = 2$, $\psi_1 = (1, 0), \psi_2 = (0, 1), \tau_i = (\frac{3}{5}, \frac{4}{5})$, with $i$ being the only component in $\mathcal{A}_1$. Then one can show that $\mathcal{B}$ and $\mathcal{C}$ satisfying the above claim do not exist.

### C.2    GAPS ON THE PROOF OF LEMMA 3.5, TERM B IN RUDI ET AL. (2013)

The proof for the assumption C can be summarized into the following three steps:

(i) $\|(\Sigma + tI)^{\frac{1}{2}}(\widehat{\Sigma} + tI)^{-\frac{1}{2}}\|^2_{\mathcal{L}^\infty(\mathcal{H})} \geq 2/3$ holds with high probability;

(ii) If $\|A^{\frac{1}{2}}B^{-\frac{1}{2}}\|^2_{\mathcal{L}^\infty(\mathcal{H})} \geq 2/3$, then $3A/2 - B$ is a semi-positive operator;

(iii) Let $t = \lambda_{\ell+1}$, $A = \Sigma + tI$, and $B = \widehat{\Sigma} + tI$.

However, the statement (ii) is not correct. For example, let $A = \begin{bmatrix} 2 & 0 & 0 \\ 0 & 2 & 0 \\ 0 & 0 & 0.1 \end{bmatrix}$ and $B = \begin{bmatrix} 3 & 0 & 0 \\ 0 & 3 & 0 \\ 0 & 0 & 0.2 \end{bmatrix}$,

then we have $\|A^{\frac{1}{2}}B^{-\frac{1}{2}}\|^2_{\mathcal{L}^\infty(\mathcal{H})} \geq 2/3$ while $3A/2 - B$ is not a semi-positive operator.

