# OpenReview forum: "Bounds on the Reconstruction Error of Kernel PCA with Interpolation Spaces Norms"
_ICLR.cc/2025/Conference — ICLR 2025 Conference Withdrawn Submission_

### Official Review · Reviewer_3t7G · 2024-11-01

**Soundness:** 4
**Presentation:** 4
**Contribution:** 4
**Rating:** 8
**Confidence:** 4

**Summary:**

The paper focuses on understanding and bounding the reconstruction error of kernel Principal Component Analysis (PCA) using interpolation space norms. This work fills gaps in previous studies on kernel PCA by providing rigorous proofs and new bounds on the reconstruction error under specific conditions. Key contributions include: 1.Upper and Lower Bounds on Reconstruction Error; 2.Applications to some interesting settings including Fixed Dimension Domain, for polynomially eigenvalue-decayed kernels and High-Dimensional Sphere, for inner-product kernels where the dimension grows along with sample size. Moreover, he paper reveals that using [\mathcal{H}]^{1}-norm in high-dimensional settings may be unsuitable due to inconsistent error behavior. In addition, a "periodic plateau" phenomenon in convergence rates is observed, where the reconstruction error rate stabilizes over certain intervals as the number of components (\ell) changes.

**Strengths:**

The most significant strength is I think overall the paper addresses a gap that exists among the most recent works in a rigorous and inspiring way. It involves both the rigorous theoretical contribution mentioned above in the summary but also provides a novel use of Interpolation Norms that I personally find helpful and interesting for technical proofs. For applications, the high-dim behavior insights mentioned above are also of great practical importance given the topic kernel PCA is a practically popular method. Besides, the paper is well distinguished from the existing works. The paper offers a thorough comparison with existing bounds, demonstrating improvements over previous results and discussing where previous work lacked rigor. This transparency about advancements and limitations strengthens the credibility of the results.

**Weaknesses:**

Overall, the paper has few weaknesses. One point can be that its results are largely theoretical, with only limited empirical validation. More comprehensive experiments across various datasets and settings would strengthen the paper by providing practical evidence to support the theoretical claims. Since after kernel PCA is a widely applied method, adding various types of empirical behavior would definitively make the paper more appealing. The other point, which would rather be some improvements, is more from a practical point of view that for example, there are parameters say $s$ in 3.4 Corollary is in practice unknown. How to do adaptation to find a data driven $n$ is also needed here as there are some works focusing on adaptation on smoothness parameters in terms of estimation and regression etc. In general, the weaknesses mainly lie in the practical side (not the main focus of the paper), which however the strengths far exceed.

**Questions:**

1. Can you please elaborator more on your paper comparing with "Rosasco L, Belkin M, De Vito E. On Learning with Integral Operators[J]. Journal of Machine Learning Research, 2010, 11(2)", where the kernel $\Sigma$ in your paper is actually an integral operator?

2. How about extending the current result to the manifold setting? As there are existing results of the convergence of some particular kernel based Laplacians on the manifolds.

3. Given similarity between MDS and kernel PCA, would the error bounds work also for MDS?

---

> ### Author Response · Authors · 2024-11-25
>
> **W**
>
> We appreciate the reviewer's thorough reading and approval of our work. For your first concern about the limited empirical validation, We have offered a simple empirical experiment in Section 4 in order to exemplify our result. However, due to the computation difficulty of experiments of kernel methods, especially under the large dimension setting, we haven't found a suitable dataset under which the experiment can be done within days.
>
> For your second concern about the choosing of parameter such as $s$, we introduced the interpolation space to bridge the gap between the two previously used norms: the RKHS norm and the $L_2(P)$ norm. The norm in the interpolation space also highlights some theoretical insights, such as the limitations of the RKHS norm in high-dimensional contexts and the periodic plateau phenomenon, which might be difficult to detect using only the RKHS norm. Hence, the adoption of interpolation space is more from the theoretical side.
>
> For your third concern about the choosing of $n$, we totally agree with you that a tighter bound of n rather than the bound $n \geq \mathcal{C}_3\ell^{2\beta}$ we give here may exist when there are some underlying properties of data. Investigating how to determine a data-driven n remains a complex issue that requires further exploration, and we may consider addressing this problem in the future. We appreciate your insightful suggestion on pursuing a data-driven approach rather than relying solely on a global bound.
>
> **Q1**
>
> Thank you for your requirement of elaborating more on the paper of Rosasco L, Belkin M, De Vito E. In fact, we utilized Proposition 10 from their paper as a lemma to support our proof of Proposition 3.4 (see Lemma B.2). Their work enabled us to verify Assumption (C) under the polynomial eigenvalue decay assumption. We have also revised our manuscript to include a citation of this literature in Section 2.2, where the properties of the integral operators are discussed.
>
> **Q2**
>
> Thank you for your insightful question on whether the current result can be extended to the manifold setting. To the best of our knowledge, the properties of specific kernels on manifolds are generally limited to a fixed-dimension context. Therefore, it might be feasible to extend the current results in Section 3.2 to manifolds. For the large dimension setting, an important lemma in our work is Lemma A.5, which shows that $ \mu_k = \Theta_d(d^{-k}) $ and $ N(d, k) = \Theta_d(d^{k}) $ for $ k \leq p+3$. Such structure in the spectrum allows us to describe reconstruction error, and can only be proved in the hypersphere setting to the best of our understanding. On the other hand, few results about the spectrum of kernels in the general domain can be found under large dimension settings (see, e.g., Remark 3.6). However, despite the difficulties of extending the large dimension results to the manifold setting, we believe that similar results shall exist, and the idea you point out is promising.
>
> **Q3**
>
> Thank you for your insightful question regarding the potential extension of our results to MDS. To the best of our knowledge, MDS typically involves a dissimilarity matrix of the samples, with elements often based on distances. The method diagonalizes this matrix and selects the subspace spanned by the leading eigenvectors. However, MDS relies heavily on the specific samples chosen, and we have not come across literature addressing MDS in a population setting, which might make the analysis more challenging than in the case of kernel PCA. Additionally, defining the interpolation space in an MDS context could be problematic. Nevertheless, we do believe that MDS might yield similar or related results due to its parallels with empirical kernel PCA, and it is of great interest and importance to derive the similar results in MDS. Please let us know if there are any misunderstandings on our part regarding MDS.

---

### Official Review · Reviewer_bkUh · 2024-11-03

**Soundness:** 3
**Presentation:** 2
**Contribution:** 3
**Rating:** 3
**Confidence:** 2

**Summary:**

This paper examines the reconstruction error of kernel Principal Component Analysis (PCA) using interpolation space norms. The authors derive upper and lower bounds for the reconstruction error of empirical kernel PCA under specific conditions. They apply these bounds to two scenarios: polynomial-eigenvalue decayed kernels in a fixed-dimension domain, and the inner product kernel on a high-dimensional sphere, comparing their bounds to existing results. Notably, this work establishes a lower bound on the sample size necessary to ensure that the empirical reconstruction error approximates the optimal reconstruction error accurately. Additionally, the authors conclude that the $H^1$-norm is unsuitable for large-dimensional settings.

**Strengths:**

The paper provides a solid theoretical analysis of kernel PCA within the framework of a generalized norm, referred to here as the interpolation space norm.

**Weaknesses:**

The presentation is suboptimal, making the paper challenging to read in its current form. There are multiple instances where notations or concepts are referenced before they are formally defined, impeding the reader’s ability to verify the correctness of the claims.

**Questions:**

- In page 1, notation $\otimes_H$ appears without definition. Also it is unclear whether $f(X)$ represents a vector or a matrix. Please clarify this notation and provide definitions for these terms.

- In page 3 there are discussions about the interpolation space norm, but this norm has not yet been defined. It’s difficult to follow the paper with references to undefined terms. Furthermore, a motivating explanation in the introduction about the significance of the interpolation space norm would be helpful. Why is this norm important, and how does it enhance the understanding of kernel PCA?

- What specific norm is used in equation (2)? Is it Frobenius norm?

- The presentation of this proposition could be improved. It references "condition (2.11) in Reiß & Wahl (2020)," yet does not restate the condition. Including the condition here would make the proposition more self-contained.

- In remark 2.3, what is meant by H norm?

- On page 5, inclusion map is not defined.

- In line 301 it is written $\langle \lambda_i^{(s-1)/2} \phi_i, \lambda_j^{(s-1)/2} \phi_i \rangle_{[H]^s} = \delta_{ij}$. Is this a definition of this inner product or is it deduced from some other fact or property? For example shouldn't the right hand side be $\lambda_i^{s-1} \delta_{ij}$ instead?

---

> ### Author Response · Authors · 2024-11-25
>
> **W**
>
> We are sorry for not making our presentation clearer. We will aim to improve clarity in the revision. However, due to the complexity of kernel PCA and the limited number of pages in an ICLR manuscript, the use of some common notations might be unavoidable. To reduce potential miscommunication, please allow us to provide a more readable summary below:
>
> **Why $ [\mathcal{H}]^{s} $ norm**
>
> Reconstruction error is often used to evaluate the performance of PCA and KPCA. For PCA, the definition is
> $$
> R(\beta_1,\cdots,\beta_{\ell}):=\mathbb{E}_ {X \sim P} \left\| X-\sum_{i=1}^{\ell}(X^\mathsf{T}\beta_i)\beta_i\right\|_ {2} ^ {2}.
> $$
> For KPCA, the definition is
> $$
> \mathcal{R}\left( \psi_{1},\ldots,\psi_{\ell} \right)
> :=
> \mathbb{E}_ {X \sim P} \left\|k(\cdot, X)-\Pi\left(\psi_1,\ldots,\psi_{\ell}\right)k(\cdot,X)\right\|^2,
> $$
> where
> $$
> \Pi(\psi_1,\ldots,\psi_{\ell}):=\sum_{i=1}^{\ell}\left\langle \cdot, \psi_i\right\rangle_{\mathcal{H}}\psi_i.
> $$
> Since kernel PCA operates in the RKHS (a function space derived by a kernel function) rather than $ \mathbb{R}^d $, various norms can be used to define the reconstruction error of KPCA. Previous works consider the $ \mathcal{H} $ norm (which is the original norm of RKHS) and the $ L_2(P) $ norm. Recent studies in RKHS regression strongly suggest that measuring the reconstruction error through $ [\mathcal{H}]^{s} $-norm would provide new insights, linking the two norms used in previous literature ($ s=0 $ refers to the $ L_2(P) $ norm; $s=1 $ refers to the $ \mathcal{H} $ norm).
>
> **Existing results:**
>
> **Lower bound of reconstruction error**
> Existing results for the lower bound of reconstruction error mainly focus on the $ \mathcal{H} $ norm and $ L_2(P) $ norm. For the $ \mathcal{H} $ norm, the proof can be simply derived by extending similar results of PCA to KPCA. (Note that when extending PCA to KPCA, the Frobenius norm becomes the RKHS norm.) For the $ L_2(P) $ norm, there are some serious gaps in the previous proof that cannot be easily fixed. (See Remark 2.6 and Appendix C.1 for details.)
>
> **Upper bound of reconstruction error**
> For the $ \mathcal{H} $ norm, the upper bound can be derived from the results in PCA. (See Proposition 2.2 for details.) Previous results of the upper bound under the $ L_2(P) $ norm are incorrect as they wrongly claimed that assumption (C) in Proposition 3.1 holds with high probability. (See the discussion below Proposition 3.1 and Appendix C.2 for details.)
>
> **Convergence rate of reconstruction error under the polynomial eigendecay setting**
> Previous literature shows that under this setting, the convergence rate is $ \ell^{-\beta+1} $ for the $ \mathcal{H} $ norm case; $ \ell^{-2\beta+1} $ for the $ L_2(P) $ norm case. Here $ \beta $ is the eigenvalue decay rate, see Assumption 3.3 for details.
>
> **Our contributions:**
>
> **Lower bound of reconstruction error**
> We establish a lower bound for the reconstruction error under the interpolation space norm. The lower bound is attained by proving that the eigenfunctions of the $ \ell $ leading eigenvalues minimize the reconstruction error (Theorem 2.5). To the best of our knowledge, we are the first to provide a rigorous proof of such theorems, especially in the interpolation space norm case.
>
> **Upper bound of reconstruction error**
> We successfully derive the upper bound of the reconstruction error under the interpolation space norm by emphasizing the importance of assumption (C) (Theorem 3.1). We also make efforts to ensure that assumption (C) can be verified in different cases. (See Remark 3.2 for details.)
>
> **Convergence rate of reconstruction error under the polynomial eigendecay setting**
> By applying our Theorem 3.1 to the polynomial eigendecay setting, we derive the convergence rate of $ \ell^{-(2-s)\beta+1} $, consistent with previous results.
>
> **Convergence rate under the large dimension setting (hypersphere)**
> We consider a hypersphere inner product kernel (a commonly used kernel in high-dimensional cases). Under this setting, we derive the convergence rate of the reconstruction error (Theorem 3.8). This leads to two notable observations. **First**, when $ s=1 $, the reconstruction error does not converge, indicating that the RKHS norm is unsuitable as an error metric in high-dimensional cases. **Second**, we identify a periodic plateau phenomenon, where the reconstruction error rate remains constant over certain ranges of $ \ell $ and decreases sharply over others (as illustrated in Figure 1(c)). A similar periodic plateau has been observed in other high-dimensional kernel methods.
>
> We hope that the above summary convinces you of our work’s significant contribution to the community and may help you reevaluate our paper. Please let us know if you have any further questions.

---

> ### Author Response · Authors · 2024-11-25
>
> **Q1**
>
> Thank you for your concern about the definition. $\otimes_{H}$ is defined in the notation part in page 4. For your convenience, we represent it here again. $a\otimes_\mathcal{H}a=\langle a, \cdot\rangle_\mathcal{H} a$, where $\langle\cdot,\cdot\rangle_\mathcal{H}$ means the inner product in space $\mathcal{H}$. In this circumstance, we provided another interpretation just behind $\otimes_{H}$. We shall change $=$ to $:=$ to avoid misunderstanding. For your second request about $f(X)$, $f(X)$ is a random variable since $X$ is an element of $(\mathcal{X},P)$ (note that $X$ and $\mathbf{X}$ are different).
>
> **Q2**
>
> Thank you for your feedback. We understand your concern that the temporarily undefined terms might make it challenging to follow the paper. In response, we indicated the sections where you can find detailed definitions. We believe it is more appropriate to present the definition of the interpolation space norm in the preliminaries section rather than in the introduction, as including all the detailed definitions upfront could make the introduction overly dense and harder to follow. The purpose of the introduction is to provide an overview and to set the context and motivation for the work.
>
> For your second concern about the significance of the interpolation space norm, as is written in our contribution and the summary above, the interpolation space norm links the two previous types of reconstruction error considered by other literature, the RKHS norm reconstruction error and the $L_2(P)$ error. Also, considering the interpolation space norm also brings us some new insights, such as the two observations found in the large dimension case, the unsuitableness of RKHS norm and the periodic plateau phenomenon.
>
> **Q3**
>
> Thank you for your question on the norm used in equation (2). As is shown in the notation part, the norm in equation (2) is the 2-norm in $R^d$. (From definitions in (2), both $X$ and $\beta_i$'s are vectors in $R^d$)
>
> **Q4**
>
> Thank you for your request for the condition mentioned in related works. We have already dded the condition in the manuscript. (All the major changes of the main manuscript have been marked blue. Typos and other minor changes are also updated in the new manuscript.) For your convenience, we display it here.
>
> **Condition** For all $s\leq \ell$, the following inequality holds:
> $$
> \frac{\lambda_s}{\lambda_s-\lambda_{\ell+1}} \sum_{j \leq s} \frac{\lambda_j}{\lambda_j-\lambda_{\ell+1}} \leq n /\left(16 C_3^2\right)
> $$
>
> **Q5**
>
> Thank you for the request for a explanation of the $\mathcal{H}$ norm in Remark 2.3. The $\mathcal{H}$ norm is the norm of the RKHS. We mention the $\mathcal{H}$ norm here before the RKHS part so as to announce that we want to compare this result with our result. We have added the explanation of $\mathcal{H}$ norm in the remark according to your advice. Also, we added a reference in the manuscript where you may find detailed properties and explanations of RKHS.
>
> Reference:A. Caponnetto and E. De Vito. Optimal rates for regularized least-squares algorithm. Foundations
> of Computational Mathematics, 7:331–368, 2007.
>
> **Q6**
>
> Thank you for pointing out the need for clarification regarding the inclusion map on page 5. The inclusion map is a standard mathematical concept defined as follows: Given a subset $A$ of $B$, the inclusion map $I$ assigns element $x$ of $A$ to $x$, the latter $x$ is treated as an element of $B$,
> $
> I:A\rightarrow B, I(x)=x.
> $
>
> **Q7**
>
> Thank you for your question. This is a definition of the inner product of the interpolation space, and we rewrite it in our revised manuscript as follows:
> $$
> \langle \lambda_i^{(s-1)/2}\phi_i, \lambda_j^{(s-1)/2}\phi_j \rangle_{[\mathcal{H}]^s} :=\delta_{ij}.
> $$
> Notice that ${\phi_i}$ is a basis of the interpolation space, hence giving the definition of the inner product between $\phi_i$ and $\phi_j$ is enough to induce the inner product and norm of the whole interpolation space. The example you give is actually the inner product under the RKHS, which is a special case of the interpolation space when $s=1$.

---

### Official Review · Reviewer_CWKU · 2024-11-03

**Soundness:** 2
**Presentation:** 4
**Contribution:** 3
**Rating:** 6
**Confidence:** 2

**Summary:**

This work is concerned with kernel PCA, and specifically with the statistical performance of the empirical estimator, defined as the $\ell$ principal components of the empirical kernel matrix $\widehat{\Sigma}_{ij} = k(X_i, X_j)$.
The main contributions are upper bounds on the empirical estimator's reconstruction error as a function of $\ell$, for several error metrics, and under several statistical settings.

The error metrics considered are the interpolation space norms $[H]^s$ for $0 \leq s \leq 1$, defined in Section 2.4. For $s=0$ this amounts to considering the estimator's $L^2$ reconstruction error, and for $s=1$ it amounts to the RKHS-norm reconstruction error.

There are three statistical settings considered:
- An abstract setting (section 3.1) where the only assumption is a condition referred to as "assumption $(C)$" in the paper. The following subsections make use of this abstract result.
- The classical setting (section 3.2) where dimension $d$ is constant and where the kernel $k$ has polynomially decaying eigenvalues.
- A high-dimensional setting (section 3.3) where sample size $n \asymp d^\gamma$ for some fixed $\gamma>1$.

Another contribution is a rigorous proof that the minimum $[H]^s$-norm reconstruction error admits a simplified expression for all $0 \leq s \leq 1$ (Theorem 2.5).

A secondary contribution is the remark that, in the high-dimensional setting of section 3.3, the RKHS-norm reconstruction error _of any estimator_ does not vanish as $n \to \infty$, implying that this error metric is unsuitable in this setting. Moreover, high-dimensional kernel PCA is shown to exhibit a similar phenomenon as in high-dimensional kernel regression: the periodic plateau behavior.

**Strengths:**

I am unable to assess the originality of this work w.r.t related literature, as I am not sufficiently familiar with this literature.

Identifying the important role played by "assumption (C)" in statistical analyses of kernel PCA is an interesting point.

The paper is very well structured and easy to follow.

**Weaknesses:**

In the proof of Theorem 2.5, I don't understand why the orthonormality constraint $(\psi_1, ..., \psi_\ell) \in B_\ell$ does not appear in the stationarity condition of $\mathcal{R}_s$, equation (6). In fact I did not manage to recover equation (6) at all. This step of the proof deserves more details.

The last sentence of the abstract is not very clear, and "$[H]^1$ norm" is not defined at that stage. Perhaps a more precise statement would be that "the RKHS norm is not a relevant error metric in high dimensions".

The figures are difficult to read, the paper would greatly benefit from making them bigger. (E.g with matplotlib, reduce figsize and increase dpi.)

The paper contains many typos and unusual wordings:
- line 17, in the abstract, remove space after opening parenthesis
- line 118, add "In", or use "contain"
- line 343, the statement of assumption (C), remove "If"
- line 376, replace "interested" by "interesting"
- throughout, consider using the word "setting" in place of "circumstance", which is less commonly used
- line 424, remove space after opening parenthesis
- line 470, add "A" at the beginning of the sentence
- line 531, replace "decayed" by "decaying", and "provide" by "provided"
- line 537, replace "on" by "of"

**Questions:**

- Please address the missing step in the proof of Theorem 2.5 (see Weaknesses).
- In Corollary 3.4, what is the dependency of the constant $C_3$ on problem parameters? is it polynomial?

---

> ### Author Response · Authors · 2024-11-25
>
> **W1: In the proof of Theorem 2.5, I don't understand why the orthonormality constraint $ (\psi_1,\ldots,\psi_{\ell})\in B_{\ell} $ does not appear in the stationarity condition of $ \mathcal{R}_{s} $, equation (6). In fact, I did not manage to recover equation (6) at all. This step of the proof deserves more details.**
>
> Thank you for your request for more details on the proof of Theorem 2.5. We have provided a proof that possesses higher readability. We hope that the updated version addresses your concerns about the proof of Theorem 2.5.
>
> **W2: The last sentence of the abstract is not very clear, and "$[\mathcal{H}]_1$ norm" is not defined at that stage. Perhaps a more precise statement would be that "the RKHS norm is not a relevant error metric in high dimensions".**
>
> Thank you for your advice on the last sentence of the abstract. We have already changed the expression according to your suggestion.
>
> **W3: The figures are difficult to read; the paper would greatly benefit from making them bigger. (E.g., with matplotlib, reduce figsize and increase dpi.)**
>
> Thank you for your kind suggestion. Following your advice, we have increased the font size of Figure 1 and the figure size of Figure 2 in our revised manuscript. We hope this makes the figures clearer.
>
> **W4: The paper contains many typos and unusual wordings.**
>
> Thank you for pointing out the typos and unusual wordings. We have revised our manuscript according to your suggestion.
>
> **Q1: Please address the missing step in the proof of Theorem 2.5 (see Weaknesses).**
>
> Thank you for requesting further explanation of the proof of Theorem 2.5. We have already provided a proof with higher readability. Please let us know if you have additional questions about the proof.
>
> **Q2: In Corollary 3.4, what is the dependency of the constant $ \mathcal{C}_3 $ on problem parameters? Is it polynomial?**
>
> Thank you for your request to explicitly express **$ \mathcal{C}_3$** . By Lemma B.2, we have **$ \sup _ {j \geq 1}|\lambda _ j-\widehat{\lambda} _ j| \leq 2\kappa \sqrt{2\tau} n^{-1/2} $**, thus **$ \widehat {\lambda}_{\ell+1}$**$ \leq 2\lambda_{\ell+1} $ holds when $ 2\kappa \sqrt{2\tau} n^{-1/2}\leq c _ {\beta} (\ell+1)^{-\beta} \leq \lambda _ {\ell+1} $. Hence, we can choose $ \mathcal{C} _ 3 = 2^{2\beta}8\kappa^2\tau c _ \beta^{-2} $.

---

> ### Comment · Reviewer_CWKU · 2024-11-27
>
> Thank you for your answer. Regarding Q2, perhaps it could be useful to future readers to remark that $C_3$ is polynomial in the parameters (except beta, but the bound can be rewritten as $C'_3 (2\ell)^{2\beta}$), and especially in $\tau$.
>
> I still have questions on the proof of Theorem 2.5, notably the case $0<s<1$ (which is the main case of interest):
> - What is meant by "maximizing $R_s(\psi_{l+1}, ...)$" on line 859?
> - Why does the equation on line 866 imply that $A$ must be block-diagonal?
> - What is meant by "$A_1$ is non-zero only on $\ell$ $e_q$'s" on line 868?
>
> By the way I noticed some typos in the updated proof:
> - on lines 792-800, the summation $q=1$ (or $p=1$) should be replaced by $q \in N$ (or $p$)
> - on line 802, $\sum_j a_{pj} a_{qj} = \delta_{pq}$, not $1$
> - on line 840, the last two factors of the left hand side, $A_1$ and $\Lambda$, are switched. This also impacts the reasoning for the case $s=1$ start on line 842.
> - on line 851, it could be helpful to explain explicitly why the space spanned by $A_1$ is invariant by the operator $\Lambda H \Lambda$ (IIUC it's as a consequence of (6)).
> - on line 852, it could also be helpful to point out explicitly that this operator is symmetric, so that the space spanned by $A_2$ is invariant.

---

### Official Review · Reviewer_aCWt · 2024-11-08

**Soundness:** 3
**Presentation:** 3
**Contribution:** 4
**Rating:** 8
**Confidence:** 3

**Summary:**

The paper gives bounds on the recontruction error of kernel PCA, measured in the full scale of interpolation spaces $[H]^s$ between $L_2$ ($s=0$) and the RKHS ($s=1$).
Analogous results exist in the literature for $s=0$ and $s=1$. However, the paper identifies a gap in the existing proofs for $s=0$ and gives an alternative, correct proof. Moreover, the results for $0<s<1$ are completely novel up to my knowledge, and they correspond to the existing ones in the limiting cases.

**Strengths:**

- The topic is of current interest, and the results are clearly presented, discussed, and placed in the context of the existing literature.
- The new bounds extend the existing results to any $0\leq s\leq 1$. The extension is significant and relevant for applications.
- The identification and correction of a bug in the result $s=0$ is interesting and relevant to the community.

**Weaknesses:**

- The results in large dimensions hold on the sphere, as is clearly explained in Section 3.3. and especially in Theorem 3.8. This is a reasonable setting, but it should be made clear upfront in the introduction (e.g., in Table 1 and in Section 1.2 under "Convergence rate of empirical kernel PCA in large dimensions"). It would also be interesting to know more precisely what are technical limitations beyond the sphere?
- The results are in part motivated by filling gaps in the existing literature, namely those discussed in Appendix C1 and Appendix C2. To the best of my understanding, these gaps are identified correctly. But the claim is quite significant, and it should be better discussed in the main text. Also, according to the two appendices, the existing arguments fail for very specific corner cases: An effort should be made to clarify if these are cases of general interest.
- There are results in the approximation theory literature that seem to be closely related and should be discussed, e.g. Theorem 3 in [1] seems to prove a version Theorem 2.5 for s=0. See also [2].


[1] G. Santin and R. Schaback, Approximation of eigenfunctions in kernel-based spaces, Adv. Comput. Math. (2016)

[2] I. Steinwart, A short note on the comparison of interpolation widths, entropy numbers, and Kolmogorov widths, J. Approx Theory (2016)

**Questions:**

Besides the points discussed above, there are the following minor points:
- Several constants are used in the introduction (Section 1) without being introduced. This makes the discussion sometimes difficult to follow.
- What does condition (2.11) in Reiss and Wahl (2020), quoted in Proposition 2.2, mean?
- Figure 1: The experimental setup is missing here, and it's unclear whether the plots correspond to an actual experiment. This should be specified.

---

> ### Author Response · Authors · 2024-11-25
>
> **W1**
>
> Thank you for your advice that the hypersphere setting should be clarified in the introduction. We have updated the introduction according to your suggestion.
>
> As for your second question, we consider the sphere setting for two main reasons:
>
> - On the one hand, the spectral properties of inner product kernels for uniform data distributed on a large-dimensional sphere are clear. In Lemma A.5 we have shown that $\mu_k = \Theta_d(d^{-k}) $ and $ N(d, k) = \Theta_d(d^{k})$ for $ k \leq p+3 $. Such a strong block structure in the spectrum, as described, leads to the periodic plateau behavior of the reconstruction error, and can only be proved in the hypersphere setting to the best of our understanding.
>
> - On the other hand, few results about the spectrum of kernels in the general domain can be found under large dimension settings (see, e.g., Remark 3.6).
>
> We will try to generalize our results to kernels in general domains in future work.
>
> **W2**
>
> Thank you for your request on more discussion of the gaps. We shall add the following discussion in the main text. However, due to the constraint of main text length, we are unable to provide the whole Appendix C in the main text. We hope that the discussion shall relieve your concern on the more discussion of gaps.
>
> - For Theorem 2.5 (which refers to Appendix C.1 for the gap), the previous gaps mainly focus on the wrong proof of the similar theorems, while the theorem itself remains true. The gap is mainly due to the wrong decomposition of some operators and sets. We give the correct proof of the theorem, and hence fill the gaps in the existing literature.
>
> - For Proposition 3.1 (which refers to Appendix C.2 for the gap), they claimed that assumption (C) holds with high probability. The gap in their proof is mainly due to the wrong claim that a specific operator is positive semi-definite.
>
> For your second concern that the arguments fail for specific cases, we guess that it is the assumption (C) that you think only fails for very specific corner cases. However, as shown in Remark 3.2, previous works consider the U-statistics, whose assumption (C) is hard to be verified. In order to overcome such difficulties, we consider a different setting so that the verification of assumption (C) becomes possible. Hence, we have made efforts so that assumption (C) can be verified in some important cases, rather than just pointing out a corner case under which the previous literature exists gaps.
>
> **W3**
>
> Thank you for your advice on the discussion of related works. After reading the literature you provided, we find that G. Santin and R. Schaback provide a link between the space spanned by the $ \mathcal{H} $ eigenfunctions and the space spanned by the $L_2 $ eigenfunctions. The literature you provided is closely related, and we have added the discussion in our work accordingly.
>
> **Q1**
>
> Thank you for your suggestion of adding the introduction of constants before using them in Section 1. We guess that the constants you mention are $ \mathcal{N}_{\Sigma}(t) $, which is a coefficient that can be bounded by no more than $ O(1/n) $. We have already added the explanation in the table. Please let us know if you have other questions about the constants used. (All the major changes of the main manuscript have been marked in blue. Typos and other minor changes are also updated in the new manuscript.)
>
> **Q2**
>
> Thank you for your request to explicitly exhibit the condition. We have already added the condition in our revised manuscript. For your convenience, we also display it here.
>
> **Condition:** For all $ s \leq \ell $, the following inequality holds:
> $$
> \frac{\lambda_s}{\lambda_s - \lambda_{\ell+1}} \sum_{j \leq s} \frac{\lambda_j}{\lambda_j - \lambda_{\ell+1}} \leq \frac{n}{16 C_3^2}
> $$
>
> **Q3**
>
> Thank you for your question considering whether Figure 1 corresponds to an experiment. Figure 1 is only a graphical illustration of the theoretical results, not an experiment. The experimental part is in Section 4. We will clarify in our manuscript that Figure 1 is only an illustration according to your suggestion.

---

> > ### Comment · Reviewer_aCWt · 2024-11-26
> > **Response to the authors**
> >
> > I would like to thank the authors for providing detailed clarifications and responses to my questions.
> > I confirm that I will keep my score to an 8.

---

> > > ### Author Response · Authors · 2024-11-26
> > >
> > > Thank you again for your important suggestions and approving the importance of our work and contribution to the community.  Please let us know if you have further questions.

---

### Note · Authors · 2024-12-02

I have read and agree with the venue's withdrawal policy on behalf of myself and my co-authors.